# E-Forcing: Improving Autoregressive Models by Treating it as an energy-based one

## Abstract

Autoregressive generative models are commonly used to solve tasks involving sequential data. They have, however, been plagued by a slew of inherent flaws due to the intrinsic characteristics of chain-style conditional modeling (*e.g.*, exposure bias or lack of long-range coherence), severely limiting their ability to model distributions properly. In this paper, we propose a unique method termed E-Forcing for training autoregressive generative models that takes advantage of a well-designed energy-based learning objective. By leveraging the extra degree of freedom of the softmax operation, we are allowed to make the autoregressive model itself an energy-based model for measuring the likelihood of input without introducing any extra parameters. Furthermore, we show that with the help of E-Forcing, we can alleviate the above flaws for autoregressive models. Extensive empirical results, covering numerous benchmarks demonstrate the effectiveness of the proposed approach.

## 1 Introduction

By factorizing the joint distribution into the product of a series of conditional distributions, autoregressive generative models (abbr. ARGMs) (Vaswani et al., 2017; Dai et al., 2019; van den Oord et al., 2016a;b; Salimans et al., 2017; Chen et al., 2018) simplify the difficult challenge of modeling high-dimensional joint distributions. They can be trained efficiently via maximum likelihood and generate samples of exceptional quality, making this technique popular for modeling distributions, especially for sequential data. Nonetheless, despite their potency and flexibility, and huge success, ARGMs still have inherent weaknesses due to the intrinsic characteristics of chain-style conditional modeling, especially when the training data is less diverse [1]. For example, ARGMs usually suffer from a discrepancy in distributions of input contexts between the training and inference stages, which causes a consequent performance drop, *i.e.*, Exposure Bias (Ranzato et al., 2016; Bengio et al., 2015). Besides, due to the nature of the greedy selection of beam search approximations, the decoded results from ARGMs may also lack long-range coherence (Deng et al., 2020).

Earlier work, both heuristic and theoretical, has been proposed to address these concerns. For instance, the exposure bias problem of ARGMs can be alleviated to some extent with scheduled sampling (Bengio et al., 2015; Mihaylova & Martins, 2019), by mixing input contexts from both real data and autoregressive generation, during the training stage. However, this scheme introduces some new problems like the over-correcting (Zhang et al., 2019) issue. In addition, at the inference stage, sampling methods such as beam search is employed to generate diverse candidates with high likelihoods, improving the quality of generated sequences. Nevertheless, these approaches result in only marginal improvements in temporal coherence.

In this paper, we propose an elegant solution, *i.e.*, E-Forcing, for the above problems of ARGMs by leveraging a deep connection between ARGMs and Energy-based models (EBMs). EBMs are a popular class of generative models that have demonstrated their effectiveness in modeling high-dimensional distributions in a variety of machine learning applications, without requiring the transformation of the target distribution into a product of conditional distributions (Zhao et al., 2017;

---

[1]When trained on massive datasets under which the underlying distribution is diverse enough, such as in large language models, this problem can be relieved because the training data covers a lot of corner cases making the model much harder to go off distribution.

Arbel et al., 2021; Gao et al., 2021). As a result, several studies (Deng et al., 2020; Bakhtin et al., 2021; Durkan & Nash, 2019) have made their attempts to benefit ARGMs from the advantages of EBMs. However, though some positive results were obtained, the existing works preferred a two-stage optimization, which first obtained a well-trained ARGM and then trained an additional EBM based on it. Such an optimization strategy not only introduced a heavy training process for EBM but also did not enable ARGMs themselves to benefit from the properties of EBM in modeling the joint distribution in a temporally more coherent way, and required more training parameters to estimate energy scores, burdening the intricacy of the learning task.

Our method of combing ARGMs and EBMs takes a different approach, which seamlessly integrates energy-based models into autoregressive models by utilizing the extra degree of freedom within the final softmax layer of the model. We show that in this way the ARGM can be trained using an energy-based learning objective, which allows the ARGM to avoid those intrinsic concerns, such as exposure bias, with the help of energy-based models as former work did (Deng et al., 2020; Bakhtin et al., 2021) whilst being free of increasing the learning model's complexity. This property makes our E-Forcing rather easy to be applied in the training process of any ARGM for any specific task, as no structural changes are required.

Besides, we follow the predominant approach for training explicit density generative models to minimize the KL divergence between the (empirical) data distribution and model distribution, which gives rise to the gradient-based contrastive divergence (CD) methods (Hinton, 2002; Kim & Bengio, 2016) for energy-based models. Typically, these methods require a Markov Chain Monte Carlo (MCMC) process to sample data from the EBM for the "negative phase" gradient estimation, which is extremely time-consuming and, meanwhile, inapplicable for discrete data, such as text. To solve this, we present a way to estimate those "negative phase" gradients through those samples generated with the network's autoregressive view instead of the EBM view, making the training feasible. Since our method combines the EBM and ARGM seamlessly as a whole, *i.e.*, the ARGM is also an EBM itself, the exposure bias problem can be mitigated due to the fact that autoregressively sampled data is involved in the "negative phase" of CD methods. On top of it, unlike ARGMs, which factor the joint distribution into a product of conditional distributions, EBMs are able to model the joint distribution directly and score each input at the sequence level instead of at the token level, which makes them capable of modeling long-range coherence.

In summary, the following contributions are made to this paper: i) We introduce a novel scheme by integrating the EBM view into autoregressive generative models seamlessly; ii) We proposed a novel method, named E-Forcing, for efficiently optimizing the energy-based autoregressive model via contrastive divergence based on importance sampling but not MCMC; iii) We successfully decrease the inherent flaws of autoregressive models — exposure bias and weak temporal coherence — by leveraging E-Forcing's two-phase optimization, which makes use of both real and generated data; iv) We demonstrate clear improvements of the proposed methods on various tasks such as language modeling, machine translation, and image generation.

## 2 BACKGROUND AND RELATED WORKS

### 2.1 ENERGY-BASED MODELS

Let $p_d$ denote the data distribution. Energy-based models (LeCun et al., 2006) are interested in learning an unnormalized energy function $\mathbf{E}_\theta(\mathbf{x})$ that defines the density(mass) function $\pi_\theta(\mathbf{x})$ as

$$\pi_\theta(\mathbf{x}) = \frac{\exp(-\mathbf{E}_\theta(\mathbf{x}))}{\mathbf{Z}_\theta}, \tag{1}$$

where $E_\theta : \mathcal{X} \to \mathbb{R}$ denotes an energy function which aims to map a data sample from data space $\mathcal{X}$ to an energy scalar, and $\mathbf{Z}(\theta) = \sum_{\mathbf{x}} \exp(-\mathbf{E}_\theta(\mathbf{x}))$ denotes the normalizing constant, also known as the partition function, which can be barely estimated. Any function can be used as an energy function to represent an EBM as long as it can generate a single scalar given some input $\mathbf{x}$ and the normalizing constant is finite[2]. Contrastive divergence algorithms are commonly used to optimize EBMs via maximum log-likelihood (Hinton, 2002; Kim & Bengio, 2016; Grathwohl et al., 2020).

---

[2]Without constraining the parametrization of $\mathbf{E}_\theta$, this can be achieved by bounding the region of space in which $x$ takes its allowed values.

Correspondingly, the gradient of the log-likelihood, which needs to be maximized, with respect to $\theta$ can be expressed as

$$\nabla_\theta \mathbb{E}_{p_d(\mathbf{x})}\Big[\log \pi_\theta(\mathbf{x})\Big] = \mathbb{E}_{\pi_\theta(\mathbf{x})}\Big[\nabla_\theta \mathbf{E}_\theta(\mathbf{x})\Big] - \mathbb{E}_{p_d(\mathbf{x})}\Big[\nabla_\theta \mathbf{E}_\theta(\mathbf{x})\Big]. \tag{2}$$

The first term on the right-hand side of Eq.2 is usually called the "negative phase" term while the second term is called the "positive phase" term.

In general, due to the challenge of sampling from EBMs, training EBMs by contrastive divergence methods (Hinton, 2002; Kim & Bengio, 2016; Grathwohl et al., 2021) is difficult, especially on high-dimensional data. MCMC methods (Nijkamp et al., 2019; Du & Mordatch, 2019; Grathwohl et al., 2020) are usually adopted for data sampling. However, these methods require enormous extra computing overheads and are not applicable when the input is discrete such as text sequences (Deng et al., 2020). As a result, a variety of recent works attempt to explore the strategy of training an EBM without MCMC. In particular, Bakhtin et al. (2021); Xu et al. (2021); Gao et al. (2020) optimize the EBMs by using noise contrastive estimation (NCE) (Gutmann & Hyvärinen, 2010; Ma & Collins, 2018). Durkan & Nash (2019) estimate the intractable normalization component by utilizing ARGMs and importance sampling. Bengio et al.; Che et al. (2020); Wang et al. (2021) skirt the challenge of collecting data in the high-dimensional data space by performing sampling using a carefully crafted latent space, which improves sampling efficiency.

## 2.2 MODELING DISTRIBUTIONS AUTOREGRESSIVELY

Modeling high-dimensional data distributions directly is usually a rather challenging task due to "the curse of dimensionality" (Bellman, 1954). One alternative method is to sequential the random variables and then factorize the joint probability distribution into the product of conditionals based on the sequence structure, which is the core idea of autoregressive generative models (ARGMs). ARGMs have been very successful, in particular for sequential data. For example, ARGMs have been widely used in language modeling (Vaswani et al., 2017; Dai et al., 2019; Radford et al., 2019), audio synthesis (van den Oord et al., 2016a), and even image generation (van den Oord et al., 2016c;b; Salimans et al., 2017).

However, the advantages of ARGMs are balanced to some extent by issues of (1) exposure bias (Ranzato et al., 2016; Bengio et al., 2015; Song et al., 2020), due to the discrepancy in input context distributions between the training and inference stages, and (2) weak long-range coherence (Deng et al., 2020), due to the inherent greedy selection of one token at a time without look-ahead.

## 2.3 THE MIXTURE OF EBMS AND GENERATIVE MODELS

The seminal idea of combing a generative model and an energy-based model has been explored by a plethora of great works (Pang et al., 2020; Durkan & Nash, 2019; Xie et al., 2019; 2020; Xiao et al., 2021; Bakhtin et al., 2021; Che et al., 2020; Arbel et al., 2021; Deng et al., 2020; Bakhtin et al., 2021; Durkan & Nash, 2019). In particular, Pang et al. (2020) aimed to learn an energy-based model (EBM) in the latent space of a generator model, so that the EBM can act as a prior model on the generator model's top-down network. VAEBM, a symbiotic composition of a variational auto-encoder and an EBM, was proposed by (Xiao et al., 2021). Arbel et al. (2021) proposed a novel training method for a GAN/EBM combined model by leveraging the Donsker-Varadham representation of KL-divergence.

Among these works, Residual EBM (Deng et al., 2020; Bakhtin et al., 2021; Durkan & Nash, 2019) and EBR (Naskar et al., 2020) may be the most related works to our paper. Authors of these works have made their attempt to benefit ARGMs from the advantages of EBMs. However, different from our work, these works utilize a two-stage optimization scheme, which first obtained a well-trained generative model and then trained an additional EBM on top of it. Such an optimization strategy does not enable ARGMs themselves to benefit from the properties of EBM in modeling the joint distribution. Besides, in order to benefit from the EBM, complicated re-sampling or re-ranking schemes are needed during inference time. It also increases parameters since it uses independent networks to represent the ARGM and the EBM, burdening the intricacy of the learning task. In contrast, we introduce the EBM inside the ARGM, treating the ARGM directly as an EBM itself.

## 3 TREATING THE ARGM AS AN EBM

In this section, we present the overall framework of our E-Forcing method for training better autoregressive models. Let $(\mathbf{x}_1, \ldots, \mathbf{x}_K)$ be a random sequence of length $K$ drawn from the real data distribution $p_d$, $\mathbf{x}_k$ denote the random variable at time step $k$, and $\mathbf{x}_{<k}$ represent the random subsequence before time step $k$, *i.e.* $\mathbf{x}_{<k} = (\mathbf{x}_1, \mathbf{x}_2, \ldots, \mathbf{x}_{k-1})$. The general spirit of our design is to model the joint distribution $p_d(\mathbf{x}_k, \mathbf{x}_{<k})$ by integrating an EBM inside the autoregressive model $q_\theta$.

Formally, given an autoregressive model $q_\theta(\mathbf{x}_1, \ldots, \mathbf{x}_K) = \prod_{k=1}^{K} q_\theta(\mathbf{x}_k | \mathbf{x}_{<k})$ parameterized by $\theta$, we introduce $K$ independent energy-based models $p_\theta(\mathbf{x}_k, \mathbf{x}_{<k})$ for each time step $k \leq K$, with the formulation following

$$p_\theta(\mathbf{x}_k, \mathbf{x}_{<k}) = q_\theta(\mathbf{x}_{<k}) \cdot \frac{e^{-\phi_\theta(\mathbf{x}_k, \mathbf{x}_{<k})}}{\mathbf{Z}_\theta}, \tag{3}$$

where $\mathbf{Z}_\theta$ is equal to $\mathbb{E}_{q_\theta}[\sum_{\mathbf{x}_k} e^{-\phi_\theta(\mathbf{x}_k, \mathbf{x}_{<k})}]$, indicating the normalization constant, $\phi_\theta(\cdot)$ represents the energy function. Essentially, $p_\theta(\mathbf{x}_k, \mathbf{x}_{<k})$ is a product EBM, defined as the product of $q_\theta$ and another EBM $\phi_\theta$.

### 3.1 DEFINITION OF THE ENERGY FUNCTION

We define the energy function $\phi_\theta(\mathbf{x}_k, \mathbf{x}_{<k})$ using $\mathbf{x}_k$'s corresponding component of network's output logits given the input context $\mathbf{x}_{<k}$ (*e.g.*, given a sequence "This is Friday." and assuming the corresponding index of the token "Friday" in the vocabulary is $i$, then the value of $-\phi_\theta$("Friday", "This is") is the $i$-th component of the output logit, namely, the input tensor of the final softmax layer).

The rationale behind such a design of energy function is out of the extra degree of freedom concealed inside the softmax transformation $\mathcal{S} : \mathbb{R}^M \to (0, 1)^M$, which can convert an unnormalized vector with size $M$ into a probability distribution consisting of $M$ probabilities

$$\mathcal{S}([z_1, \ldots, z_M]) = [\frac{e^{z_1}}{\sum_{i=1}^{M} e^{z_i}}, \ldots, \frac{e^{z_M}}{\sum_{i=1}^{M} e^{z_i}}]. \tag{4}$$

It's easy to observe that the softmax operation is unaffected by the input vector's overall magnitude, that is, $\mathcal{S}([z_1, \ldots, z_M]) = \mathcal{S}([z_1, \ldots, z_M] + C), \forall C \in \mathbb{R}$. Such a property allows us to model the energy function by using the ARGM itself instead of introducing a new network.

### 3.2 ENERGY-BASED LEARNING OBJECTIVE

Other than making the $q_\theta$ to match $p_d$, E-forcing has an additional training objective to make the $K$ parametric distributions $p_\theta(\mathbf{x}_k, \mathbf{x}_{<k})$ to match the real data distribution $p_d(\mathbf{x}_k, \mathbf{x}_{<k})$ at any time step $k \leq K$. This can be achieved by minimizing the Kullback-Leibler (KL) divergence between the distributions for each time step of a sequence,

$$\mathbf{D}_{KL}\Big(p_d(\mathbf{x}_k, \mathbf{x}_{<k}) || p_\theta(\mathbf{x}_k, \mathbf{x}_{<k})\Big), \forall k \in [1, K], \tag{5}$$

We attempt to use contrastive divergence methods (Hinton et al., 1995; Kim & Bengio, 2016) to minimize the objective 5 by descending the gradient w.r.t. $\theta$ according to Eq. 2 for each time step. Specifically, given an arbitrary time step $k$, we have the corresponding gradient of objective 5 with respect to $\theta$

$$\nabla_\theta \mathcal{L}_{EBM-CD} = \underbrace{\mathbb{E}_{p_d}\Big[\nabla_\theta \mathbf{E}_\theta(\mathbf{x}_k, \mathbf{x}_{<k})\Big]}_{\text{Positive Phase Gradient}} - \underbrace{\mathbb{E}_{p_\theta}\Big[\nabla_\theta \mathbf{E}_\theta(\mathbf{x}_k, \mathbf{x}_{<k})\Big]}_{\text{Negative Phase Gradient}}. \tag{6}$$

where $\mathbf{E}_\theta(\mathbf{x}_k, \mathbf{x}_{<k}) = \phi_\theta(\mathbf{x}_k, \mathbf{x}_{<k}) - \log q_\theta(\mathbf{x}_{<k})$.

Optimization via Eq. 6 involves sampling data from the model distribution $p_\theta$ and can thus lead to the discovery of non-data-like samples, whose likelihood is then explicitly reduced as the corresponding energy increases during the training. E-Forcing is therefore not plagued by the exposure bias problem naturally. Besides, because we model the joint distribution at each time step, E-Forcing can assess

the sequence up to the current time step as a whole and generate more coherent data using energy sampling (Deng et al., 2020). However, the negative phase gradient is frustrating to compute, especially for discrete data (*e.g.* text) where common MCMC methods (Welling & Teh, 2011) can not even be applied. Therefore, we propose a novel variant of contrastive divergence methods for E-Forcing's optimization in Section 4.

## 4 OPTIMIZATION

The key obstacle of optimizing the objective 5 via contrastive divergence methods (Hinton, 2002)(*i.e.* descends the gradient of Eq. 6) is sampling data from the model distribution $p_\theta$ for estimating the negative phase gradient. The common MCMC algorithms are not desirable for generating "negative" samples because they are rather time-consuming, and not applicable to discrete data. In order to make the optimization process both efficient and feasible, we modified the original CD methods by means of the importance sampling technique (Horvitz & Thompson, 1952), which holds two parts of gradient estimation.

### 4.1 POSITIVE PHASE GRADIENTS

Since the training set consists of *i.i.d.* samples sampled from the real distribution $p_d$, the computing of positive phase gradients is not difficult. To be specific, by replacing $\mathbf{E}_\theta(\mathbf{x}_k, \mathbf{x}_{<k})$ with the form $\phi_\theta(\mathbf{x}_k, \mathbf{x}_{<k}) - \log q_\theta(\mathbf{x}_{<k})$ in Eq.6, the positive phase gradient $\mathcal{G}_+^{(k)}(\theta)$ with respect to parameter $\theta$ can be written into

$$\mathcal{G}_+^{(k)}(\theta) = \mathbb{E}_{p_d}\Big[\nabla_\theta \phi_\theta(\mathbf{x}_k, \mathbf{x}_{<k}) - \nabla_\theta \log q_\theta(\mathbf{x}_{<k})\Big]. \tag{7}$$

Since carrying out sample estimation of the expectation over the data distribution $p_d$ is viable, and the score $\phi_\theta(\mathbf{x}_k, \mathbf{x}_{<k})$ can be acquired by simply accessing the output logit of ARGM (according to the definition of $\phi_\theta$ in Sec. 3), the first term of the positive phase gradient $\mathcal{G}_+^{(k)}$ can likewise be readily computed. Besides, we can observe that the second term $\mathbb{E}_{p_d}[-\nabla_\theta \log q_\theta(\mathbf{x}_{<k})]$ of $\mathcal{G}_+^{(k)}(\theta)$ is the negative gradient of likelihood $q_\theta(\mathbf{x}_{<k})$'s logarithm, which is exactly the objective of maximizing the autoregressive generative model $q_\theta$'s log-likelihood.

### 4.2 NEGATIVE PHASE GRADIENTS

The estimation of negative phase gradients $\mathcal{G}_-^{(k)}(\theta) = \mathbb{E}_{p_\theta}[\nabla_\theta \phi_\theta(\mathbf{x}_k, \mathbf{x}_{<k}) - \nabla_\theta \log q_\theta(\mathbf{x}_{<k})]$, on the other hand, is more involved. Sampling data from $p_\theta$ is required for estimating the expectation $\mathbb{E}_{p_\theta}$, whereas $p_\theta$ is the introduced energy-based autoregressive model, which is an explicit autoregressive generative model and we can only access its modeled density(mass) function $p_\theta$.

Inspired by the idea of importance sampling, we substitute the troublesome estimation of the expectation over distribution $p_\theta$ with the expectation over distribution $q_\theta$, which is the underlying autoregressive model that can generate samples considerably easier. Accordingly, the negative phase gradient $\mathbb{E}_{\mathbf{x}_k, \mathbf{x}_{<k} \sim p_\theta}[\nabla_\theta \mathbf{E}_\theta(\mathbf{x}_k, \mathbf{x}_{<k})]$ has the following form (See the detailed derivation in Appendix B),

$$\mathcal{G}_-^{(k)}(\theta) = \mathbb{E}_{q_\theta}\Big[\mathbf{w}(\mathbf{x}_{<k})\big[\nabla_\theta \phi_\theta(\mathbf{x}_k, \mathbf{x}_{<k}) - \nabla_\theta \log q_\theta(\mathbf{x}_{<k})\big]\Big], \tag{8}$$

$$\text{where } \mathbf{w}(\mathbf{x}_{<k}) = \frac{\sum_{\mathbf{x}_k} e^{-\phi_\theta(\mathbf{x}_k, \mathbf{x}_{<k})}}{\mathbb{E}_{q_\theta(\mathbf{x}'_{<k})}[\sum_{\mathbf{x}_k} e^{-\phi_\theta(\mathbf{x}_k, \mathbf{x}'_{<k})}]}. \tag{9}$$

According to Eq.8, all the estimated expectations only need sampling from the autoregressive model $q_\theta$ rather than the distribution $p_\theta$, and the reweighing weight $\mathbf{w}$ in Eq. 9 does not involve expectation computation over distribution $p_\theta$ either. Generally speaking, producing data from an autoregressive model is a simple ancestral sampling process and naturally suitable for discrete data, as compared with sampling straight from an explicit generative density estimator, which needs MCMC approaches (Durkan & Nash, 2019). Besides, the term $\mathbb{E}_{\mathbf{x}_{<k} \sim q_\theta(\mathbf{x}_{<k})}[\mathbf{w}(\mathbf{x}_{<k})\nabla_\theta \log q_\theta(\mathbf{x}_{<k})]$ in Eq. 8 can be regarded as a re-weighted gradient of $q_\theta$'s information entropy with respect to $\theta$. This term can be optimized similarly to the teacher-forcing training of the autoregressive model with

the "teacher" sequence generated autoregressively by the model itself. The scheduled sampling methods (Bengio et al., 2015; Ranzato et al., 2016; Mihaylova & Martins, 2019) are similar to this term but without the re-weighting factor.

Moreover, the reweighing weight $\mathbf{w}$ of Eq. 9 can be further refined (see the derivation in Appendix B.3) and we can observe that $\mathbf{w}(\mathbf{x}_{<k}) = \mu(\mathbf{x}_{<k})/\mathbb{E}_{\mathbf{x}'_{<k}}\mu(\mathbf{x}_{<k})$, where $\mu(\mathbf{x}_{<k}) = p_\theta(\mathbf{x}_{<k})/q_\theta(\mathbf{x}_{<k})$, indicating the possibility of which distribution ($p_\theta$ or $q_\theta$) the input context $\mathbf{x}_{<k}$ is most likely to come from. Correspondingly, $\mathbf{w}(\mathbf{x}_{<k})$ reflects the context $\mathbf{x}_{<k}$'s relative magnitude of $\mu(\mathbf{x}_{<k})$ compared with the average among all potential contexts—the larger the value of $\mathbf{w}(\mathbf{x}_{<k})$, the more likely the context $\mathbf{x}_{<k}$ in the data space coming from $p_\theta$, which is modeled by the product of autoregressive models and EBMs. During training, those input sequences with contexts more likely from $p_\theta$ than $q_\theta$ will be assigned larger weights $\mathbf{w}$ while others will be assigned smaller weights $\mathbf{w}$.

### 4.3 FINAL OPTIMIZATION OF E-FORCING

---
**Algorithm 1** Optimizing ARGMs with E-Forcing

---
**Given:** a training dataset $\mathcal{E} \sim p_d$ , random-initialized autoregressive model $q_\theta$, $K \in \mathbb{N}$ is the generation length

    **for** iteration $i = 1; i \leq$ max iterations; $i + 1$ **do**

        Sample minibatch $B = \{(c_i, s_i)\}_{i=1}^n \sim \mathcal{E}$         ▷ $s_i$ is of length $K$, $c_i$ is the context of $s_i$

        **if** i≤N **then**         ▷ After N iterations, we start applying E-Forcing

            $\nabla_\theta \mathcal{L}_{total} \leftarrow \sum_{k=1}^K \nabla_\theta \mathcal{L}_{AR}^{(k)}(B)$

        **else**

            Autoregressively generate $|B|$ samples from $q_\theta$ conditioned on $c_i$, denoted as $\tilde{B}$

            $\nabla_\theta \mathcal{L}_{total} \leftarrow \sum_{k=1}^K \nabla_\theta \mathcal{L}_{AR}^{(k)}(B) + \lambda_k \nabla_\theta \mathcal{L}_{EBM-CD}^{(k)}(B, \tilde{B})$

        **end if**

        Update $\theta \leftarrow \theta - \eta \nabla_\theta \mathcal{L}_{total}$         ▷ $\eta$ denotes learning rate

    **end for**

---

Finally, with the help of the above estimation of gradients regarding two phases of Eq. 6, we are able to optimize the product EBM $p_\theta$ via descending the estimated gradient of contrastive divergence loss $\nabla_\theta \mathcal{L}_{EBM-CD}$ for any time step $k$

$$\nabla_\theta \mathcal{L}_{EBM-CD}^{(k)}(\theta) = \mathcal{G}_+^{(k)}(\theta) - \mathcal{G}_-^{(k)}(\theta). \tag{10}$$

Eq. 10 can be easily estimated by using "positive" samples from the given training dataset and autoregressively generated "negative" samples from $q_\theta$.

Nevertheless, training the model from scratch with the energy-based learning objective alone can not work well in practice. The reason is simple: at the initial stage of the training process, what we have is just a randomly initialized network that can barely generate anything meaningful. This fact indicates disjoint supports between the real distribution $p_d$ and modeled distribution $p_\theta$. Importance sampling fails in this case. Hence, to make the optimization more feasible, we pre-train the entire model with an autoregressive loss $\mathcal{L}_{AR}$ by teacher-forcing for a few epochs before introducing the energy-based learning objective. In sum, the final gradient concerning parameter $\theta$ at each update iteration is

$$\nabla_\theta \mathcal{L}_{total}(\theta) = \sum_{k=1}^K \nabla_\theta \mathcal{L}_{AR}^{(k)}(\theta) + \lambda_k \nabla_\theta \mathcal{L}_{EBM-CD}^{(k)}(\theta), \tag{11}$$

where $\lambda_k$ adjusts the ratio between the two objectives for the time step $k$. The intact optimization procedure is shown in Algorithm 1[3]. We found that an exponentially descending configuration of coefficients $\lambda_k$ according to the order of time steps works well. One possible reason is that such a set of coefficients can remedy the imbalanced training signal by negative phase gradients in Eq. 8 among time steps.

---

[3]We take $K$ to be the length of a segment of the transformer.

## 5 EXPERIMENTS

To empirically corroborate the effectiveness of E-Forcing and show its broad applicability, we have conducted extensive experiments on applications, such as language modeling and machine translation. In this section, we will first introduce these experimental setups and analyze the obtained results. Besides, we also carried out a series of experiments to further show our E-Forcing method's ability to resolve ARGMs' inherent flaws(*i.e.* exposure bias and incoherence generation). More experimental settings as well as analytical experiments are shown in Appendix A and C.

### 5.1 APPLICATION TO LANGUAGE MODELING

For the language modeling task, three different datasets, WikiText-103 (Merity et al., 2017), Toronto Book Corpus (Zhu et al., 2015; Kiros et al., 2015), and CC-news (Mackenzie et al., 2020), are chosen as testbeds; two autoregressive network structures are used to evaluate the effectiveness: vanilla Transformer (Vaswani et al., 2017) ("Tr-Base" for short) and Transformer-XL (Dai et al., 2019) ("Tr-XL" for short). We regard the vanilla training with the teacher forcing technique as the baseline method. Besides, we also compared our E-Forcing with the residual EBM Deng et al. (2020) method, which is a typical method to improve the performance of language models by utilizing EBMs. In order to benefit from the EBM, the residual EBM method requires a new network to estimate the energy scores and imposes a Top-K energy resampling scheme during inference[4].

The final results are reported in Table 1. We can see from the results that E-Forcing outperforms two pure autoregressive models with clear margins over all three benchmarks. Specifically, on the Wikitext-103 benchmark, our E-Forcing improves the performance of the Transformer-Base model and Transformer-XL model by 0.62 PPL points (from 30.56 to 29.94) and 0.30 PPL points (from 24.20 to 23.90) respectively; on CC-news and Toronto Book Corpus benchmarks, our method obtains 0.51 ppl and 0.47 ppl performance gain respectively and gets further improvement once energy resampling technique was applied. Besides, though residual EBM's learning parameters are twice as ours and their method is unable to directly benefit autoregressive models without Top-K energy resampling, our E-Forcing achieves comparable results to them, even slightly better on Toronto Book Corpus and Wikitext-103 benchmarks.

| Method | PPL ↓ | | |
|---|---|---|---|
| | CC-News | Toronto Book Corpus | WikiText103 |
| **Baseline** (Tr-Base) | 18.29 | 17.57 | 30.56 |
| **Baseline** (Tr-XL) | - | - | 24.20 |
| **Residual EBM** (Tr-Base) | **15.57-15.58** | 16.98-17.00 | 29.88-29.93 |
| **Residual EBM** (Tr-XL) | - | - | 23.85-23.87 |
| **E-Forcing** (Tr-Base) | 15.78 | 17.10 | 29.94 |
| **E-Forcing** (Tr-XL) | - | - | 23.90 |
| **E-Forcing + E.R.** (Tr-Base) | 15.63-15.67 | **16.89-16.93** | 29.81-29.84 |
| **E-Forcing + E.R.** (Tr-XL) | - | - | **23.79-23.82** |

Table 1: Language modeling performance of different models on three benchmarks. Evaluation is conducted using perplexity (PPL). **E.R.** is the abbreviation of Energy Resampling technique (Bakhtin et al., 2021), which serves as a necessary module of Residual EBM.

### 5.2 APPLICATION TO NEURAL MACHINE TRANSLATION

We further evaluate E-Forcing's effectiveness on neural machine translation (NMT), which can be regarded as a conditional generation task. We mainly analyze E-Forcing on the IWSLT14 dataset, which includes six different language pairs ({German, Spanish, Italian} → English and English → {German, Spanish, Italian}) (Hereafter we abbreviate English, German, Spanish, Italian as "EN",

---

[4]It is worth noting that Top-K energy resampling can not get the PPL directly. Bakhtin et al. (2021) provides a way to approximate PPL, which leads to an estimated interval of PPL.

| Method | Label Smoothing | Scheduled Sampling | Beam Searching | BLEU ↑ | | | | | | Avg. |
|---|---|---|---|---|---|---|---|---|---|---|
| | | | | DE→EN | EN→DE | EN→IT | IT→EN | ES→EN | EN→ES | |
| Base | - | - | - | 32.44±0.06 | 26.64±0.10 | 27.92±0.03 | 30.48±0.08 | 38.61±0.11 | 35.42±0.09 | 31.92 |
| | - | - | 5 B | 33.62±0.07 | 27.41±0.08 | 28.72±0.04 | 31.39±0.05 | 39.55±0.12 | 36.38±0.07 | 32.85 |
| | ✔ | - | - | 33.68±0.03 | 27.62±0.04 | 28.81±0.07 | 31.42±0.07 | 39.85±0.13 | 36.71±0.09 | 33.02 |
| | ✔ | - | 5 B | 34.61±0.08 | 28.46±0.06 | 29.72±0.10 | 32.29±0.03 | 40.64±0.07 | 37.48±0.05 | 33.87 |
| | ✔ | ✔ | - | 34.23±0.06 | 27.96±0.03 | 29.26±0.11 | 31.93±0.08 | 40.16±0.03 | 37.21±0.04 | 33.46 |
| | ✔ | ✔ | 5 B | 35.10±0.04 | 28.73±0.04 | 29.97±0.07 | 32.64±0.12 | 40.91±0.06 | 37.93±0.10 | 34.21 |
| E-Forcing | - | - | - | 32.99±0.10 | 27.15±0.03 | 28.33±0.12 | 31.13±0.04 | 39.56±0.01 | 36.07±0.02 | 32.54 |
| | - | - | 5 B | 34.06±0.06 | 27.97±0.08 | 29.26±0.09 | 31.90 ±0.13 | 40.30±0.03 | 36.92 ±0.09 | 33.40 |
| | ✔ | - | - | 33.97 ±0.08 | 28.03±0.04 | 29.13 ±0.02 | 31.84 ±0.11 | 40.32 ±0.03 | 36.96 ±0.07 | 33.38 |
| | ✔ | - | 5 B | 34.93 ±0.05 | 28.91 ±0.12 | 30.04 ±0.11 | 32.56 ±0.04 | 41.01 ±0.06 | 37.73 ±0.12 | 34.20 |
| | ✔ | ✔ | - | 34.58 ±0.09 | 28.38 ±0.12 | 29.56 ±0.10 | 32.11 ±0.03 | 40.93 ±0.03 | 37.56 ±0.07 | 33.85 |
| | ✔ | ✔ | 5 B | **35.36** ±0.05 | **29.11** ±0.04 | **30.25** ±0.09 | **32.82** ±0.11 | **41.58** ±0.07 | **38.19** ±0.03 | **34.55** |

Table 2: Comparison of BLEU scores between our approach E-Forcing and the base ARGM trained just with cross-entropy loss on six translation pairs of IWSLT14 datasets. We use "-" to denote that the training trick is not used while "✔" indicates we use it. "**5 B**" represents we use beam searching with 5 beams.

"DE", "ES", "IT"). In addition, we also reported the result of E-Forcing over the WMT16 (English → German) benchmark, which is a relatively larger dataset, in Table 3.

Results concerning IWSLT14 are shown in Table 2, where we use a six-layer vanilla transformer as the base autoregressive model. We test not only the pure performance of E-Forcing but also the compatibility with other techniques. In detail, we can observe that (1) without any particular engineering, E-Forcing outperforms the vanilla training with cross-entropy singly(teacher-forcing) by 0.62 (31.92 → 32.54) BLEU points on average, especially on three translation pairs—38.61 → 39.56 on Spanish-to-English, 30.48 → 31.13 on Italian-to-English, 35.42 → 36.07 on English-to-Spanish. (2) E-Forcing is compatible with other techniques like scheduled sampling, which can help alleviate the exposure bias problem to some extent. They are not mutually exclusive and can work together to further improve the performance of the base ARGM. (3) However, since scheduled sampling can reduce exposure bias and beam search can somewhat alleviate the flaws caused by greedy selection at each time step, the performance gain of E-Forcing when all these tactics are combined is only 0.34 (34.21 → 34.55), which is lower than the 0.62 (31.92 → 32.54) obtained when the model is purely trained without these other techniques.

Additionally, Table 3 shows the performance of E-Forcing on the WMT16 English → German task. For two different model sizes, enabling label smoothing (L.S.) improves model performance by 0.52 and 0.35, respectively. The performance of the base transformer model further increases to 28.36 BLEU points when scheduled sampling (S.S.) is used, while the larger model improves to 29.23 points. E-Forcing paired with label smoothing and scheduled sampling yields the highest scores of 28.62 and 29.44, respectively. Overall, our training strategy outperforms ARGM's vanilla teacher-forcing training and can have uniformly favorable impacts across different models and dataset sizes.

| Model | L.S. | S.S. | E-Forcing | BLEU ↑ |
|---|---|---|---|---|
| Tr-Base | - | - | - | 27.56 |
| | ✔ | - | - | 28.04 |
| | ✔ | ✔ | - | 28.36 |
| | ✔ | ✔ | ✔ | **28.62** |
| Tr-Large | - | - | - | 28.70 |
| | ✔ | - | - | 29.05 |
| | ✔ | ✔ | - | 29.23 |
| | ✔ | ✔ | ✔ | **29.44** |

Table 3: Translation performance of proposed E-Forcing on WMT16 English→German, evaluated with BLEU. We uniformly use 5 beams when applying beam search. "**L.S.**" denotes Label Smoothing and "**S.S.**" denotes Scheduled Sampling.

## 5.3 EFFECT ON THE EXPOSURE BIAS

We follow the analytic experiments in the work (Zhang et al., 2019) to show that our E-Forcing is capable of alleviating the exposure bias problem. To be concrete, we randomly select 1K pairs from the training data for each translation pair and use the trained autoregressive model which applied E-Forcing to decode the source sentences, and then count the ground truth words whose probabilities in the predicted distributions produced by our E-Forcing are greater than those produced by the baseline and denote the number as $\mathcal{N}$. The ratio of $\mathcal{N}$ to the total number of words tested is calculated. The detailed results are shown in Table 4. We find that the results on all different tasks are greater

than 50%, which demonstrates the ability of our E-Forcing in alleviating the exposure bias problem to some extent.

| Trans. Pairs | DE→EN | EN→DE | EN→IT | IT→EN | ES→EN | EN→ES |
|---|---|---|---|---|---|---|
| $\mathcal{N}$ | 14203 | 14554 | 14976 | 13952 | 16021 | 15359 |
| **Total** | 22148 | 23057 | 23654 | 23744 | 23860 | 22775 |
| **Ratio** | 64.12% | 63.12% | 63.31% | 59.76% | 68.33% | 67.43% |

Table 4: The effect of E-Forcing on the exposure bias problem. Each test set of translation tasks contains 1K sentences selected randomly. $\mathcal{N}$ denotes the ground truth words whose probabilities in the predicted distributions produced by E-Forcing are greater than those produced by the baseline.

## 5.4 EFFECT ON THE INCOHERENCE OF GENERATION

| Translation Task | Scheduled Sampling | E-Forcing Training | Target Sentence Length | | | All Test |
|---|---|---|---|---|---|---|
| | | | [0, 25) | [25, 49) | [50, ∞) | |
| **De→En** | - | - | 37.72 ±0.04 | 33.24 ±0.06 | 30.86 ±0.07 | 34.61 ±0.08 |
| | ✔ | - | 38.20 ±0.07 | 33.76 ±0.03 | 31.08 ±0.06 | 35.10 ±0.04 |
| | ✔ | ✔ | 38.37 ±0.06 | 33.92 ±0.09 | 31.43 ±0.04 | 35.36 ±0.05 |
| **It→En** | - | - | 35.20 ±0.03 | 32.73 ±0.02 | 26.86 ±0.05 | 32.29 ±0.03 |
| | ✔ | - | 35.52 ±0.09 | 33.25 ±0.08 | 26.95 ±0.14 | 32.64 ±0.12 |
| | ✔ | ✔ | 35.56 ±0.10 | 33.33 ±0.13 | 27.21 ±0.07 | 32.82 ±0.11 |
| **Es→En** | - | - | 43.37 ±0.05 | 39.67 ±0.08 | 37.14 ±0.06 | 40.64 ±0.07 |
| | ✔ | - | 43.61 ±0.09 | 40.00 ±0.04 | 37.38 ±0.06 | 40.91 ±0.06 |
| | ✔ | ✔ | 43.84 ±0.10 | 40.35 ±0.05 | 38.07 ±0.04 | 41.58 ±0.07 |

Table 5: Performance comparison on the IWSLT14 test set for the different lengths of sentences on three translation tasks (German to English, Italian to English, and Spanish to English). Performance is evaluated by the BLEU score.

We also attempted to quantitatively validate that our E-Forcing can benefit ARGMs by improving the coherence of generation. Table 5 shows the BLEU scores of generated translations on the IWSLT14 test set with respect to different lengths of the source sentences. Intuitively, due to the cumulative effect of greedy selection at each time step, the collection of samples with longer sentences ought to be more plagued by the incoherence of generations problem. Our approach can outperform the vanilla training in all three length intervals, especially in the lengthy sentence interval $[50, \infty]$, indicating that our E-Forcing can resolve the incoherence problem to a degree.

## 6 CONCLUSIONS AND FUTURE WORK

In this paper, we propose a novel training method dubbed E-Forcing for ARGMs by treating them as EBMs. This is achieved by defining the energy function using the softmax operation's extra degree of freedom within an autoregressive network. We further design a unique way to improve the training of E-Forcing using importance sampling. Experimental results demonstrate the effectiveness of E-Forcing to alleviate exposure bias and incoherence problems of ARGMs. In the future, we expect to extend E-Forcing on other sequential generation tasks (*e.g.* text summarization, audio generation) and incorporate the proposed methodology into other advanced autoregressive architectures.

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

## A  EXPERIMENTAL SETTINGS

In this section, we introduce detailed setups of different benchmarks as well as the information of corresponding datasets.

### A.1  DATASETS

We conducted our experiments based on 7 datasets over different learning tasks:

1. **WikiText-103** comprises 103 million training tokens from 28 thousand articles, with an average length of 3.6 thousand tokens per article.

2. **Toronto Book Corpus** consists of fiction books in 16 different genres, totaling about half a billion words.

3. **CC-news** is a de-duplicated subset of the English portion of the CommonCrawl news dataset, which totals around 16 Billion words.

4. **IWSLT14** contains about 170k training sentence pairs, 7k valid pairs, and 7k test pairs. It has six different domains of language, and each two of them can consist of a translation pair.

5. **WMT16** contains 103M training tokens from 28K articles, with an average length of 3.6K tokens per article, which allows testing the ability of long-term dependency modeling.

6. **MNIST** is a large collection of handwritten digits. It has a training set of 60,000 examples and a test set of 10,000 examples.

7. **CIFAR-10** is a subset of the Tiny Images dataset and consists of 60000 32x32 color images. The images are labeled with one of 10 mutually exclusive classes.

### A.2  IMPLEMENTING SETUPS

| Hyper-Parameters | Translation-IWSLT14 | Translation-WMT16 | | Language Modeling | |
|---|---|---|---|---|---|
| | **Tr-Base** | **Tr-Base** | **Tr-Large** | **Tr-Base** | **Tr-XL** |
| **Number of Layers** | 12 | 12 | 12 | 6 | 16 |
| **Hidden Embed Size** | 512 | 512 | 1024 | 512 | 410 |
| **FC-Layer Embed Size** | 1024 | 2048 | 4096 | 2048 | 2100 |
| **Attention Heads** | 4 | 8 | 16 | 8 | 10 |
| **Dropout** | 0.3 | 0.3 | 0.3 | 0.1 | 0.1 |
| **Learning Rate** | 5e-4 | 1e-3 | 1e-3 | 5e-4 | 2.5e-4 |
| **lr scheduler** | inverse_sqrt | inverse_sqrt | inverse_sqrt | inverse_sqrt | cosine |
| **Warm up Updates** | 4000 | 4000 | 4000 | 4000 | 10000 |
| **Weigth Decay** | 1e-4 | 0.0 | 0.0 | 1e-2 | 0.0 |
| **E-Forcing Start Epoch** | 15 | 15 | 10 | 15 | 10 |

Table 6: Hyperparameters of different model structures and datasets. "Tr-Base", "Tr-Large", and "Tr-XL" indicate Transformer-Base, Transformer-Large, and Transformer-XL respectively

We uniformly use the Adam optimizer. The training will be stopped once the model has not obtained better performance for 20 epochs on the validation set. For translation tasks, the length of generated fake sentences, which is used for the computing of the negative phase in Eq. 10, is dependent on the source sequence whilst for language modeling tasks, we fix the length of generated fake sentences as 50 during training. The model structures for language modeling and machine translation tasks are shown in Table 6. As for the model structures of the image generation task, we use the official structure reported by PixelCNN (van den Oord et al., 2016c) and Gated PixelCNN (van den Oord et al., 2016b) without modification. The source code will be released once upon acceptance. We use the same batch of samples generated autoregressively to approximate both the expectations in Eq.10 and weight **w** (*i.e.*, shared), which does not need to sample twice. The number of samples in a batch is dynamic while the maximum number of the total tokens in a batch is fixed (4096 in our experiments). If the length of sequences in a batch is 32, then it includes 4096 / 32 = 128 samples in total. It is a common strategy in language generation tasks and has been used in many frameworks(*e.g.*

Fairseq (Ott et al., 2019)). We generate samples autoregressively as many as the number of sequences in the current batch at each update iteration.

## B    DERIVATION OF THE NEGATIVE PHASE GRADIENT

In this section, we show the detailed derivation of Eq. 8. Formally, as shown in Sec. 3, given an autoregressive model $q_\theta(\mathbf{x}_{<k}) = \prod_{l=1}^{k-1} q_\theta(\mathbf{x}_l|\mathbf{x}_{<l})$ ($k$ denotes the time step) with parameters $\theta$, we define a product of the autoregressive model and an EBM as follows

$$p_\theta(\mathbf{x}_k, \mathbf{x}_{<k}) = q_\theta(\mathbf{x}_{<k}) \cdot \frac{e^{-\phi_\theta(\mathbf{x}_k, \mathbf{x}_{<k})}}{\mathbf{Z}_\theta}, \tag{12}$$

where $q_\theta(\mathbf{x}_{<k}) = \prod_{l=1}^{k-1} q_\theta(\mathbf{x}_l|\mathbf{x}_{<l})$, $\mathbf{Z}_\theta$ is the normalization term and equal to $\mathbb{E}_{\mathbf{x}'_{<k} \sim q_\theta}[\sum_{\mathbf{x}_k} e^{-\phi_\theta(\mathbf{x}_k, \mathbf{x}'_{<k})}]$. The optimization of $p_\theta(\mathbf{x}_k, \mathbf{x}_{<k})$ includes two phases, and the gradient w.r.t $\theta$ of "negative phase" is

$$-\mathbb{E}_{\mathbf{x}_{<k} \sim p_\theta}[\nabla_\theta \log q_\theta(\mathbf{x}_{<k})] + \mathbb{E}_{\mathbf{x}_k, \mathbf{x}_{<k} \sim p_\theta}[\nabla_\theta \phi_\theta(\mathbf{x}_k, \mathbf{x}_{<k})]. \tag{13}$$

Next, we will show the specific derivation about how to transform Eq. 13 into Eq. 8.

### B.1    DERIVATION OF THE FIRST TERM

The first term $\mathbb{E}_{\mathbf{x}_{<k} \sim p_\theta}[\nabla_\theta \log q_\theta(\mathbf{x}_{<k})]$ can be processed as follows

$$\begin{aligned}
\mathbb{E}_{\mathbf{x}_{<k} \sim p_\theta}[\nabla_\theta \log q_\theta(\mathbf{x}_{<k})] &= \sum_{\mathbf{x}_{<k}} p_\theta(\mathbf{x}_{<k}) \nabla_\theta \log q_\theta(\mathbf{x}_{<k}) \\
&= \sum_{\mathbf{x}_{<k}} \sum_{\mathbf{x}_k} p_\theta(\mathbf{x}_k, \mathbf{x}_{<k}) \nabla_\theta \log q_\theta(\mathbf{x}_{<k}) \\
&= \sum_{\mathbf{x}_{<k}} q_\theta(\mathbf{x}_{<k}) \frac{\sum_{\mathbf{x}_k} e^{-\phi_\theta(\mathbf{x}_k, \mathbf{x}_{<k})}}{\mathbf{Z}_\theta} \nabla_\theta \log q_\theta(\mathbf{x}_{<k}) \\
&= \mathbb{E}_{\mathbf{x}_{<k} \sim q_\theta(\mathbf{x}_{<k})}[\mathbf{w}(\mathbf{x}_{<k}) \nabla_\theta \log q_\theta(\mathbf{x}_{<k})],
\end{aligned} \tag{14}$$

where we have $\mathbf{w}(\mathbf{x}_{<k}) = \frac{\sum_{\mathbf{x}_k} e^{-\phi(\mathbf{x}_k, \mathbf{x}_{<k})}}{\mathbb{E}_{\mathbf{x}'_{<k} \sim q_\theta(\mathbf{x}_{<k})}[\sum_{\mathbf{x}_k} e^{-\phi_\theta(\mathbf{x}_k, \mathbf{x}'_{<k})}]}$ because

$$\begin{aligned}
\mathbf{w}(\mathbf{x}_{<k}) &= \frac{\sum_{\mathbf{x}_k} e^{-\phi(\mathbf{x}_k, \mathbf{x}_{<k})}}{\mathbf{Z}_\theta} = \frac{\sum_{\mathbf{x}_k} e^{-\phi(\mathbf{x}_k, \mathbf{x}_{<k})}}{\sum_{\mathbf{x}_{<k}} \sum_{\mathbf{x}_k} q_\theta(\mathbf{x}_{<k}) e^{-\phi_\theta(\mathbf{x}_k, \mathbf{x}_{<k})}} \\
&= \frac{\sum_{\mathbf{x}_k} e^{-\phi(\mathbf{x}_k, \mathbf{x}_{<k})}}{\sum_{\mathbf{x}_{<k}} q_\theta(\mathbf{x}_{<k}) \sum_{\mathbf{x}_k} e^{-\phi_\theta(\mathbf{x}_k, \mathbf{x}_{<k})}} \\
&= \frac{\sum_{\mathbf{x}_k} e^{-\phi(\mathbf{x}_k, \mathbf{x}_{<k})}}{\mathbb{E}_{\mathbf{x}_{<k} \sim q_\theta(\mathbf{x}_{<k})}[\sum_{\mathbf{x}_k} e^{-\phi_\theta(\mathbf{x}_k, \mathbf{x}_{<k})}]}.
\end{aligned} \tag{15}$$

## B.2 DERIVATION OF THE SECOND TERM

Then, we tackle the second term $\mathbb{E}_{\mathbf{x}_k,\mathbf{x}_{<k}\sim p_\theta}[\nabla_\theta\phi_\theta(\mathbf{x}_k,\mathbf{x}_{<k})]$ as follows

$$
\begin{aligned}
\mathbb{E}_{p_\theta}\left[\nabla_\theta\phi_\theta(\mathbf{x}_k,\mathbf{x}_{<k})\right] &= \sum_{\mathbf{x}_k,\mathbf{x}_{<k}} p_\theta(\mathbf{x}_k,\mathbf{x}_{<k})\nabla_\theta\phi_\theta(\mathbf{x}_k,\mathbf{x}_{<k}) \\
&= \sum_{\mathbf{x}_k,\mathbf{x}_{<k}} p_\theta(\mathbf{x}_k,\mathbf{x}_{<k})\frac{q_\theta(\mathbf{x}_k,\mathbf{x}_{<k})}{q_\theta(\mathbf{x}_k,\mathbf{x}_{<k})}\nabla_\theta\phi_\theta(\mathbf{x}_k,\mathbf{x}_{<k}) \\
&= \sum_{\mathbf{x}_k,\mathbf{x}_{<k}} q_\theta(\mathbf{x}_k,\mathbf{x}_{<k})\frac{q_\theta(\mathbf{x}_{<k})\cdot e^{-\phi_\theta(\mathbf{x}_k,\mathbf{x}_{<k})}}{\mathbf{Z}_\theta\cdot q_\theta(\mathbf{x}_k,\mathbf{x}_{<k})}\nabla_\theta\phi_\theta(\mathbf{x}_k,\mathbf{x}_{<k}) \\
&= \mathbb{E}_{\mathbf{x}_k,\mathbf{x}_{<k}\sim q_\theta(\mathbf{x}_k,\mathbf{x}_{<k})}\left[\frac{e^{-\phi_\theta(\mathbf{x}_k,\mathbf{x}_{<k})}}{q_\theta(\mathbf{x}_k|\mathbf{x}_{<k})}\cdot\frac{1}{\mathbf{Z}_\theta}\nabla_\theta\phi_\theta(\mathbf{x}_k,\mathbf{x}_{<k})\right] \\
&= \sum_{\mathbf{x}_{<k}} q_\theta(\mathbf{x}_{<k})\sum_{\mathbf{x}_k} q_\theta(\mathbf{x}_k|\mathbf{x}_{<k})\frac{e^{-\phi_\theta(\mathbf{x}_k,\mathbf{x}_{<k})}}{q_\theta(\mathbf{x}_k|\mathbf{x}_{<k})}\cdot\frac{1}{\mathbf{Z}_\theta}\nabla_\theta\phi_\theta(\mathbf{x}_k,\mathbf{x}_{<k}) \\
&= \sum_{\mathbf{x}_{<k}} q_\theta(\mathbf{x}_{<k})\sum_{\mathbf{x}_k} e^{-\phi_\theta(\mathbf{x}_k,\mathbf{x}_{<k})}\cdot\frac{1}{\mathbf{Z}_\theta}\nabla_\theta\phi_\theta(\mathbf{x}_k,\mathbf{x}_{<k}) \\
&= \mathbb{E}_{q_\theta(\mathbf{x}_{<k})}\left[\sum_{\mathbf{x}_k}\frac{e^{-\phi_\theta(\mathbf{x}_k,\mathbf{x}_{<k})}}{\mathbf{Z}_\theta}\nabla_\theta\phi_\theta(\mathbf{x}_k,\mathbf{x}_{<k})\right] \\
&= \mathbb{E}_{q_\theta(\mathbf{x}_{<k})}\left[\sum_{\mathbf{x}_k}\frac{e^{-\phi_\theta(\mathbf{x}_k,\mathbf{x}_{<k})}}{\sum_{\mathbf{x}_k}e^{-\phi_\theta(\mathbf{x}_k,\mathbf{x}_{<k})}}\cdot\frac{\sum_{\mathbf{x}_k}e^{-\phi_\theta(\mathbf{x}_k,\mathbf{x}_{<k})}}{\mathbf{Z}_\theta}\nabla_\theta\phi_\theta(\mathbf{x}_k,\mathbf{x}_{<k})\right] \\
&= \mathbb{E}_{q_\theta(\mathbf{x}_{<k})}\left[\sum_{\mathbf{x}_k}q_\theta(\mathbf{x}_k|\mathbf{x}_{<k})\mathbf{w}(\mathbf{x}_{<k})\nabla_\theta\phi_\theta(\mathbf{x}_k,\mathbf{x}_{<k})\right] \\
&= \mathbb{E}_{q_\theta(\mathbf{x}_{<k})}\left[\mathbb{E}_{a\sim q_\theta(\mathbf{x}_k|\mathbf{x}_{<k})}[\mathbf{w}(\mathbf{x}_{<k})\nabla_\theta\phi_\theta(\mathbf{x}_k,\mathbf{x}_{<k})]\right] \\
&= \mathbb{E}_{\mathbf{x}_k,\mathbf{x}_{<k}\sim q_\theta(\mathbf{x}_k,\mathbf{x}_{<k})}[\mathbf{w}(\mathbf{x}_{<k})\nabla_\theta\phi_\theta(\mathbf{x}_k,\mathbf{x}_{<k})]
\end{aligned}
\tag{16}
$$

where $\mathbf{w}(\mathbf{x}_{<k})$ is also equal to $\frac{\sum_{\mathbf{x}_k}e^{-\phi(\mathbf{x}_k,\mathbf{x}_{<k})}}{\mathbf{Z}_\theta}$. Combining Eq. 14 and Eq. 16, we can obtain an equivalent form of the gradient of the negative phase without any expectation over $p_\theta$ as

$$
-\mathbb{E}_{\mathbf{x}_{<k}\sim q_\theta(\mathbf{x}_{<k})}[\mathbf{w}(\mathbf{x}_{<k})\nabla_\theta\log q_\theta(\mathbf{x}_{<k})] + \mathbb{E}_{\mathbf{x}_k,\mathbf{x}_{<k}\sim q_\theta(\mathbf{x}_k,\mathbf{x}_{<k})}[\mathbf{w}(\mathbf{x}_{<k})\nabla_\theta\phi_\theta(\mathbf{x}_k,\mathbf{x}_{<k})], \tag{17}
$$

$$
\textbf{where}\quad \mathbf{w}(\mathbf{x}_{<k}) = \frac{\sum_{\mathbf{x}_k}e^{-\phi(\mathbf{x}_k,\mathbf{x}_{<k})}}{\mathbb{E}_{\mathbf{x}'_{<k}\sim q_\theta(\mathbf{x}_{<k})}[\sum_{\mathbf{x}_k}e^{-\phi_\theta(\mathbf{x}_k,\mathbf{x}'_{<k})}]}. \tag{18}
$$

## B.3 FURTHER REFINEMENT OF $\mathbf{w}$

The reweighing weight $\mathbf{w}$ can be further deduced as

$$
\begin{aligned}
\mathbf{w}(\mathbf{x}_{<k}) &= \frac{\sum_{\mathbf{x}_k}e^{-\phi(\mathbf{x}_k,\mathbf{x}_{<k})}}{\mathbb{E}_{\mathbf{x}'_{<k}\sim q_\theta(\mathbf{x}_{<k})}[\sum_{\mathbf{x}_k}e^{-\phi_\theta(\mathbf{x}_k,\mathbf{x}'_{<k})}]} = \frac{\sum_{\mathbf{x}_k}\frac{p_\theta(\mathbf{x}_k,\mathbf{x}_{<k})}{q_\theta(\mathbf{x}_{<k})}}{\mathbb{E}_{\mathbf{x}'_{<k}\sim q_\theta(\mathbf{x}_{<k})}[\sum_{\mathbf{x}_k}\frac{p_\theta(\mathbf{x}_k,\mathbf{x}_{<k})}{q_\theta(\mathbf{x}_{<k})}]} \\
&= \frac{\frac{p_\theta(\mathbf{x}_{<k})}{q_\theta(\mathbf{x}_{<k})}}{\mathbb{E}_{\mathbf{x}'_{<k}\sim q_\theta(\mathbf{x}_{<k})}[\frac{p_\theta(\mathbf{x}_{<k})}{q_\theta(\mathbf{x}_{<k})}]} = \frac{\mu(\mathbf{x}_{<k})}{\mathbb{E}_{\mathbf{x}'_{<k}}\mu(\mathbf{x}_{<k})},
\end{aligned}
\tag{19}
$$

where $\mu(\mathbf{x}_{<k})$ is defined as $\frac{p_\theta(\mathbf{x}_{<k})}{\tilde{q}_\theta(\mathbf{x}_{<k})}$.

## C MORE EXPERIMENTAL ANALYSIS

### C.1 ANALYSIS TO TOP-K ENERGY RE-SAMPLING

Top-K energy re-sampling in the inference stage is introduced by Bakhtin et al. (2021), which collects many candidate sequences generated autoregressively in the inference stage and then re-samples

| Trans. Pairs | | DE→EN | EN→DE | EN→IT | IT→EN | ES→EN | EN→ES |
|---|---|---|---|---|---|---|---|
| | 0 | 34.93 | 28.91 | 30.04 | 32.56 | 41.01 | 37.73 |
| $k$ | 5 | **34.97** | 28.92 | **30.08** | **32.60** | **41.07** | 37.71 |
| | 10 | 34.95 | **28.95** | 30.07 | 32.59 | 41.03 | **37.75** |

Table 7: The effect of Top-K correction in the inference stage. We tested BLEU scores of using different $k$ on different translation pairs of IWSLT14 dataset.

from them depending on their energy scores estimated by the network. To measure the effectiveness of the Top-K energy re-sampling towards our method, we conduct an ablation study on neural machine translation task by selecting different K = {0, 5, 10}. The results are shown in Table 7 and performances are evaluated by using the BLEU score. From Table 7, we observe that the benefits brought by Top-K sampling is minor (K={5, 10}), when compared with the model without Top-K sampling (K=0). This together with the results shown in Table 1 shows that our E-Forcing can considerably benefit the base autoregressive model even without the energy resampling technique.

## C.2 APPLICATION TO IMAGE GENERATION

In order to illustrate the effectiveness and generality of our method in processing different modality tasks, we further show the results of applying E-Forcing to image generation in this section. We apply E-Forcing to Pixel-CNN (Van Oord et al., 2016) and its variant Gated Pixel-CNN (Oord et al., 2016). Experiments are carried out on the MNIST and CIFAR-10 datasets.

Table 8 summarizes the quantitative results measured by per-pixel negative log-likelihood (NLL). We can see that with the help of our E-Forcing, both the Pixel-CNN and the Gated Pixel-CNN can obtain improvements in all datasets (0.17 → 0.15 and 3.14 → 3.07 for Pixel-CNN on MNIST and CIFAR10 respectively and 0.14 → 0.12 and 3.03 → 2.97 for Gated Pixel-CNN on MNIST and CIFAR10 respectively). This is further evidence in favor of the energy-based learning objective for improving autoregressive models.

| Model | Test (Train) NLL ↓ | |
|---|---|---|
| | MNIST | CIFAR-10 |
| **Pixel-CNN** | 0.17 (0.13) | 3.14 (3.08) |
| **Pixel-CNN (w/E-Forcing)** | **0.15 (0.12)** | **3.07 (2.98)** |
| **Gated Pixel-CNN** | 0.14 (0.11) | 3.03 (2.90) |
| **Gated Pixel-CNN (w/E-Forcing)** | **0.12 (0.10)** | **2.97 (2.87)** |

Table 8: Performance of E-Forcing with different base networks on MNIST and CIFAR-10 in bits/dim (lower is better), training performance in brackets.

## C.3 THE EFFECT OF DIFFERENT START EPOCHS OF E-FORCING

In addition, we have studied the effect of different start epochs of E-Forcing on the performance of language modeling, which can be seen in Table 9. From this, we may deduce that starting E-Forcing training at the 15th and 10th epoch yields the best results for Transformer-Base and Transformer-XL respectively, whereas starting earlier or later yields a performance decline. It is reasonable because, if E-

| Model | Start Epoch of E-Forcing | | | | |
|---|---|---|---|---|---|
| Structure | 5 | 10 | 15 | 20 | 25 |
| **Tr-Base** | 30.38 | 30.12 | **29.94** | 30.05 | 30.29 |
| **Tr-XL** | 24.12 | **23.90** | 23.96 | 24.05 | 24.16 |

Table 9: Exploring the effect of different start epochs of E-Forcing on Wikitext103 benchmark. Performances are evaluated by perplexity (PPL).

Forcing was introduced too early, the autoregressive model may not have been optimized well at that moment. As a result, the quality of generation for the "negative phase" would be terrible, making energy-based training unstable. On the other hand, the underlying autoregressive model can be modified only marginally if E-Forcing was introduced when the ARGM training is virtually complete.

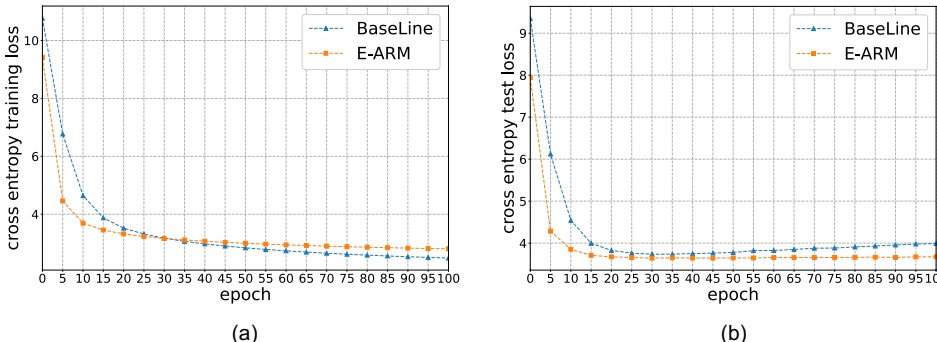

Figure 1: (a) Cross entropy loss curves on IWSLT14 Spanish to English translation task on training set. The blue and orange colors represent base model and E-Forcing respectively; (b) Cross entropy loss curves on IWSLT14 Spanish → English translation task on test set.

## C.4 ANALYSIS TO MODEL'S CONVERGENCE

In this section, we will investigate the convergence of our E-Forcing. To begin, we first train a base Transformer model ("Tr-Base" architecture shown in Table 6) on the IWSLT14 Spanish to English training set for baseline and E-Forcing method respectively, and then record the training loss and test loss (in cross-entropy) at the end of each epoch. The loss curves are plotted in Figure 1. From Figure 1, we can see that (1) at the start of the training, our E-Forcing converges slightly faster than the baseline. (2) As the training process progresses, the cross entropy of the baseline on the training set will gradually decrease, at a faster rate than E-Forcing. On the other hand, the test loss curve of the baseline will fall initially and then slowly rise after 50 epochs while E-Forcing always remains stable convergence. This phenomenon also shows that our E-Forcing method can effectively prevent over-fitting so that obtaining better generalization.

## C.5 ABLATION STUDY WITH DIFFERENT ARCHITECTURE CHOICES

| Training Methods | GRU | LSTM | ENAS | DEQ | Tr-XL | TNet |
|:---:|:---:|:---:|:---:|:---:|:---:|:---:|
| Teacher-Forcing | 92.48 | 78.93 | 58.60 | 57.10 | 54.55 | 54.19 |
| E-Forcing | 90.12 | 76.97 | 56.89 | 55.55 | 53.49 | 53.24 |

Table 10: The ablation study of E-forcing over different choices of the architecture of AR models with the comparison of vanilla teacher-forcing training. We tested PPL scores using different AR models on the Penn Treebank dataset

In this section, we conducted an ablation study to investigate our E-Forcing model's generalization ability over different sequential models. We tested over 6 different sequential models, which are GRU (Chung et al., 2014), LSTM (Hochreiter & Schmidhuber, 1997), ENAS (Pham et al., 2018), TrelisNet(TNET for short) (Bai et al., 2019b), DEQ (Bai et al., 2019a) and Transformer-XL (Dai et al., 2019) on Penn Treebank (Marcus et al., 1993) dataset, which is a relatively small dataset and widely used in machine learning for NLP (Natural Language Processing) research. In general, we can observe that our E-Forcing can achieve improvement over all base AR models applied, which indicates it is a universally applicable training method for AR models.

## C.6 CASES STUDIES

To better understand the advantages of our method in correcting error tokens, we also prepare some translation cases in IWSLT14 German → English, as shown in Table 11.

| Source Sentence(German) | Predicted Target Sentence(English) |
|---|---|
| wenn ich ihnen 600 zeitschriften zeige und sie in 10 kategorien aufteile oder ich ihnen 400 zeitschriften zeige, und diese in 20 kategorien aufteile, dann glauben sie, dass ich ihnen mehr auswahl und eine bessere auswahlerfahrung gegeben habe, als ich ihnen die 400 gegeben hätte gegenüber dem, wenn ich ihnen die 600 gegeben hätte. | **GroundTruth**: if i show you 600 magazines and i divide them up into 10 categories, versus i show you 400 magazines and divide them up into 20 categories, you believe that i have given you more choice and a better choosing experience if i gave you the 400 than if i gave you the 600.

**Baseline**: if i show you 600 magazines and i split them in 10 categories, or i'm showing them 400 magazines, and i'm going to split them up into 20 categories, you think i've given them more choices and better choice than i would have given them the 400 over the time that i gave them the 600.
**Baseline + S.S.**: if i show you 600 magazines and i give you 400 magazines in 10 categories, and i give you 400 magazines, and i can split them up in 20 categories, then you think i've given you more choice and a better selection than i would have given you the 400 of which if i gave you the 600.
**Ours**: if i show you 600 magazines and i divide them into 10 categories, or i show you 400 magazines, and i divide them into 20 categories, you think i've given you more choices and better selection experience than i gave you the 400 of whom if i gave you the 600. |
| und ich weiß definitiv, dass es für mich – in meiner situation – sehr gefährlich wäre, anzufangen, diesen dunklen pfad der vermutung sozusagen herunterzu-sickern – besonders in dem umstand, in dem ich mich in meiner karriere gerade befinde. | **GroundTruth**: and i definitely know that, in my case – in my situation – it would be very dangerous for me to start sort of leaking down that dark path of assumption, particularly given the circumstance that i'm in right now in my career.
**Baseline**: and i know definitely, for me, it would be very dangerous to begin to do this dark path of suspect – especially in the circumstance that i'm in my career right now.
**Baseline + S.S.**: and i know definitely it would be – in my situation – very dangerous to start, to kind of settle down this dark path of presumption – especially in the circumstance in which i'm in my career right now.
**Ours**: and i definitely know that it's for me – in my situation – very danger-ous to start to sickle down this dark path of suspicion, in particular, in the circumstance of where i'm in my career right now. |
| wir haben das licht ausgeschaltet, legten es in ein vakuum und saugten die ganze luft aus und kühlten es bis fast zum jetzt, ganz alleine im aufzug, war das stück metall frei, sich zu verhalten wie immer es wollte. | **GroundTruth**: we turned off the lights, and then we put it in a vacuum and sucked out all the air, and then we cooled it down now, all alone in the elevator, the little chunk of metal is free to act however it wanted.
**Baseline**: we turned the light off, put it in a vacuum and sucked it out all the air and cooled it up until almost now, all the way alone, the piece of metal was open to behave as it was.
**Baseline + S.S.**: we turned the lights off, we put it into a vacuum, and we sucked all the air, and we cooled it all the way up to now, all over the place, the piece of metal was free to behave whatever it wanted.
**Ours**: we turned off the lights, we put it into a vacuum and we sucked all the air out, and we cooled it up until almost now, all alone in the elevator, the piece of metal was free to behave whatever it wanted. |
| und im grunde können sie das betrachten, wissen sie, als eine tyrannei des erin-nernden selbst, und sie können sich das erinnernde selbst denken als eins, das sozusagen das erlebende selbst schleppt durch erfahrungen, die das erlebende selbst nicht braucht. | **GroundTruth**: and basically you can look at this, you know, as a tyranny of the remembering self, and you can think of the remembering self sort of dragging the experiencing self through experiences that the experiencing self doesn't need.
**Baseline**: and basically, you can think of this, you know, as a tyranny of self, and you can think of the memorable self as one that kind of weaves the living self through experiences that don't need the life itself.
**Baseline + S.S.**: and basically, you can look at this, you know, as a tyrannei of memorial self, and you can think of the memorial self as one that kind of sucks the living self through experiences that don't need the living self.
**Ours**: and basically, you can look at that, you know, as a tyranny of the remembering self, and you can think of the memory itself as one, which is sort of dragging the living self through experiences that the living self doesn't need. |
| wir sind an der schwelle zu erstaunlichen, erstaunlichen ereignissen auf vielen gebieten. und doch denke ich wirklich, dass wir hunderte, 300 jahre vor die aufklärung zurück gehen müssten, um eine zeit zu finden, in der wir fortschritt bekämpft haben, in der wir über diese dinge heftiger getritten haben, an mehr fronten als jetzt. | **GroundTruth**: we're on the verge of amazing, amazing events in many fields, and yet i actually think we'd have to go back hundreds, 300 years, before the enlightenment, to find a time when we battled progress, when we fought about these things more vigorously, on more fronts, than we do now.

**Baseline**: we are at the threshold of amazing, amazing events in many areas, and yet i really think that we have to go back hundreds and 300 years before the enlightenment to find a time when we have fought progress in which we have driven more of these things than now.
**Baseline + S.S.**: we're at the threshold of amazing, amazing events in many areas. and yet, i really think that we have to go back hundreds and hundreds of years before the enlightenment to find a time when we have struggled with progress in which we have driven on these things more powerful, more fronts than now.
**Ours**: we're at the threshold to amazing, amazing events in many areas, and yet i really think that we have to go back hundreds and 300 years before the enlightenment to find a time when we fought progress, where we've been fighting about these things to more fronts than we have now. |

Table 11: Translation cases on IWSLT14 De→En test set, generated by the baseline method, baseline with scheduled sampling and our E-Forcing. The italic font means the mismatch translation

## C.7 EVALUATION WITH OTHER METRICS

| Trans. Pairs | Scheduled Sampling | E-Forcing Training | Metrics | | | | |
|---|---|---|---|---|---|---|---|
| | | | ROUGE-1 ↑ | ROUGE-2↑ | ROUGE-L↑ | METEOR↑ | BLEU↑ |
| **De → En** | - | - | 66.51 | 43.69 | 63.69 | 64.35 | 34.61 |
| | ✔ | - | 66.83 | 44.08 | 64.02 | 64.61 | 35.10 |
| | ✔ | ✔ | **67.46** | **44.77** | **64.78** | **65.13** | **35.36** |
| **It → En** | - | - | 64.50 | 40.65 | 61.69 | 62.18 | 32.29 |
| | ✔ | - | 64.73 | 40.97 | 61.94 | 62.51 | 32.64 |
| | ✔ | ✔ | **65.27** | **41.51** | **62.49** | **62.80** | **32.82** |
| **Es → En** | - | - | 71.10 | 49.47 | 68.78 | 68.94 | 40.64 |
| | ✔ | - | 71.36 | 49.53 | 68.96 | 69.28 | 40.91 |
| | ✔ | ✔ | **71.91** | **50.17** | **69.65** | **69.63** | **41.58** |

Table 12: Comparison of ROUGE-1, ROUGE-2, ROUGE-L, METEOR, and BLEU scores between our approach E-Forcing and the base ARGM trained just with cross-entropy loss on three translation pairs of IWSLT14 datasets. The value is expressed in percentage. We use "Tr-Base" as the network architecture.

To further evaluate the effectiveness of our proposed E-Forcing, we also evaluate our method by using other metrics, such as ROUGE Lin (2004) and METEOR Banerjee & Lavie (2005) for neural machine translation. The results are shown in Table 12. In Table 12, the improvements of E-Forcing in different metrics is consistent with the conclusion of Table 2, which further prove the effectiveness of our E-Forcing method.

