# OpenReview forum: "E-Forcing: Improving Autoregressive Models by Treating it as an Energy-Based One"
_ICLR.cc/2023/Conference — Submitted to ICLR 2023_

### Official Review · Reviewer_W653 · 2022-10-20

**Confidence:** 5
**Correctness:** 3
**Technical Novelty And Significance:** 3
**Empirical Novelty And Significance:** 2
**Recommendation:** 5

**Clarity, Quality, Novelty And Reproducibility:**

The writing is clear and of good quality.
The novelty is mediocre, but the method could be useful.
The experiments are reproducible quite well.

**Strength And Weaknesses:**

Pros: Interesting idea, extensive experimental evaluation against the standard baseline, correct derivations.
Cons: The method is a minor extension upon existing techniques; the authors do not explain how more computationally demanding this method is compared to the baselines, and whether these extra costs are worth the extra accuracy.

**Summary Of The Paper:**

The paper proposes a method for training autoregressive generative models in a way that casts these models into Energy-Based models. In essence, the method uses the logits from the Categorical output distribution of an autoregressive model to define an energy function; this is the basis for the subsequent treatment. As maximization of the energy function entails a contrastive divergence type of sampling, which may prove ineffective, the authors provide an importance sampling-based variant.


**Summary Of The Review:**

The method is interesting, and the recasting of the categorical output distribution into the potential function of an energy-based model is nice. The training algorithm entails contributions of some merit.

On the other hand, the experiments are not convincing at all; they do not elaborate on computational costs required for the extra accuracy, and do not perform an ablation study with different architecture choices which may affect performance.

---

> ### Author Response · Authors · 2022-11-15
> **Response to Reviewer W653, part[2/2]**
>
> - **Training Stage**: We admit that the training speed would be affected to some extent compared to the vanilla training via teacher forcing since our E-Forcing has to generate “fake” samples autoregressively during the optimization procedure. However, different implementation tricks applied and hardware types used will largely influence the final training time, and different researchers, who hold different numbers of computing resources, hold a distinct tolerance towards the training time required. In general, our method requires about 2.4 times over translation tasks(IWSLT14 dataset) and 3.1 times over language modeling tasks(Wiki103 dataset) than the training time of original AR training (teacher-forcing). Well, it is true that our E-Forcing is not that efficient at the training stage, however, if we compare it with other Energy-based methods, things might be much more tolerable. For example, JEM[5] trained 36 hours on the simple CIFAR 10 dataset, while the vanilla training of a classifier usually costs no more than 3 hours(The main reason is due to the rather time-consuming MCMC procedure during each training iteration), Residual EBM and EBR[1, 2] requires no less than 2 times training time compared to vanilla training since they require to train two independent but same size networks, and it can not be accelerated via concurrently training these two networks since their training of EBM require a well-trained AR model so that the training has to carry out sequentially.
>
>     There are several parts of our method, which can influence the final efficiency of training. The first one is the max-length allowed for generated sequences during the training via the contrastive divergence method, while the second one is the start-epoch of using E-Forcing training. Setting different max-lengths of generated sequences and different start-epoch of using E-Forcing loss can introduce trade-offs between training efficiency and performance (Please refer to Appendix A.2 for detailed setting over different benchmarks).
>
>
> We think this training and inference efficiency of our method is tolerable, since, in most realistic cases, people care more about inference efficiency instead of training efficiency, especially for those works based on EBMs, which are usually notorious for training efficiency.
>
> Besides, one possible way to accelerate the training efficiency of our methods is to incorporate the idea shown in the work of Xu et al [5], who proposed a unique way that uses autoregressive models in the latent space followed by a decoder that decodes the autoregressively generated latent feature into the original data space, instead of constructing an autoregressive model in the data space. If we can move the autoregressive procedure from the data level to the latent feature level, our E-Forcing training efficiency can correspondingly improve since those samples generated in the latent space are enough in this case.
>
> [**An ablation study with different architecture choices**]
>
> Thanks for the advice, we indeed reported five different architecture results in total over all experiments: Transformer-base(Tr-Base), Transformer-large(Tr-large), Transformer-XL(Tr-XL) as well as PixelCNN and Gated Pixel-CNN, which are reported in Appendix(All detailed information about these three structures can be referred to in Appendix A.2). However, we admit that the most experiments in the main body were tested based on transformer architectures, which might result in your concern about the number and kind of architectures tested. To this end, we add a corresponding ablation study with 5 more architecture choices(GRU, LSTM, TCN, TrelisNet, efficient Nas) on Penn Treebank dataset to our latest version, please refer to Appendix C.5. In general, we can observe that our method can benefit all architectures which are tested.
>
>
> [1] Yuntian Deng, Anton Bakhtin, Myle Ott, Arthur Szlam, and Marc’Aurelio Ranzato. Residual energy-based models for text generation. ICLR 2020.
>
> [2] Bhattacharyya S, Rooshenas A, Naskar S, Sun S, Iyyer M, McCallum A. Energy-based reranking: Improving neural machine translation using energy-based models. ACL 2021.
>
> [3] Will Grathwohl, Kuan-Chieh Wang, Jorn-Henrik Jacobsen, David Duvenaud, Mohammad Norouzi, and Kevin Swersky. Your classifier is secretly an energy based model and you should treat it like one. ICLR 2020.
>
> [4] Conor Durkan and Charlie Nash. Autoregressive energy machines. ICML 2019.
>
> [5] Yilun Xu, Yang Song, Sahaj Garg, Linyuan Gong, Rui Shu, Aditya Grover, and Stefano Ermon. Anytime sampling for autoregressive models via ordered autoencoding. ICLR 2021

---

> ### Author Response · Authors · 2022-11-15
> **Response to Reviewer W653, part[1/2]**
>
> Dear Reviewer W653, we are grateful for your valuable comments. Below are our responses. If you have any problems with our response or have more questions about our work, it is welcome to give more comments. We will try our best to give our response as soon as possible.
>
> [**The method is a minor extension upon existing techniques**]
>
> Since the specific existing techniques mentioned are not listed in the comments, we elaborate the difference between our E-Forcing work and other methods which perhaps looks similar to our work (but actually have essential difference) as far as we know, as follows
>
> - **The re-ranker (re-sampler) based method, such as Residual EBM[1] and EBR[2]**: Our method is designed with a fundamentally different goal. The main goal of our model is to improve the training of autoregressive models. The EBR method and Residual EBM method, however, treat the AR model as fixed and train an independent energy-based model $E_\theta$ to reweight the samples. Although such re-ranker-based(re-sampler) methods can improve the performance, the test efficiency will be sacrificed during inference since every prediction needs an extra procedure for re-sampling. In contrast, our method’s design is orthogonal with the Re-Ranker-based methods: E-Forcing is a kind of energy-based training method for the AR model, and the goal of E-forcing is to improve the AR model itself via energy-based training. In other words, we view the AR model itself as an EBM while EBR, as well as Residual EBM, used EBM as a re-ranker apart from the AR model.
>
>     Besides, EBR is designed especially for translation tasks while Residual EBM is designed for language modeling, which will introduce the inductive bias towards these special tasks and the performance of these models to other tasks remain unclear. Our method, however, is designed as a novel training method for AR models, which, theoretically, can be applied by AR models to various tasks. Moreover, the EBR method used an extra pre-trained network which introduced knowledge outside the target dataset.
>
> - **JEM**[3]: The data and the EBM formulation are different from JEM. JEM only works on continuous data and classification tasks. In order to make energy-based models work on discrete AR models, a series of techniques are introduced in our model (product EBM, importance sampling, etc). Besides, the time-consuming MCMC methods are required by their method during their optimization, while our optimization procedure only needs sample data autoregressively, which is relatively more efficient compared to MCMC methods.
> - **AEM**[4]: AEM also attempts to combine the AR model and EBM, however, their goals are entirely different from ours. AEM introduce the AR model to facilitate the training of EBMs. That is, they focus on getting a better *EBM* with the help of the AR model since they opine that it is easier to obtain reliable estimates of normalizing constants in low dimensions by decomposing the density estimation task in an autoregressive manner. In contrast, we focus on improving *AR models* via the proposed energy-based training mechanism. Besides, the detailed framework is also different. Similar to Residual EBM[1] and EBR[2], they also regard the AR model and EBM as two independent modules while we regard the AR model as an EBM.
>
> [**Computational Efficiency**]
>
> Thanks for the comments; the computation efficiency can be discussed with respect to the training stage and inference stage, respectively. Generally speaking, our E-Forcing can benefit the AR model even without additional overheads in the inference stage, which is a very good property since people usually care more about efficiency during deploying, though it is true that it needs more computation resources during training.
>
> - **Inference Stage**: One of the merit of our E-Forcing is that it can benefit the AR model *even without additional overhead in the inference stage* since it regards the AR model as an EBM and training it follows an energy-based objective via CD methods, which mitigates the problems, such as exposure bias and lack of long-range coherence(please refer section 3 and section 4 in our paper). In contrast, other energy-based methods[1, 2] which utilized the EBM as a re-ranker(re-sampler) inevitably introduce additional overheads not only in the training stage(They need to train an independent EBM apart from the trained AR model) but also in the inference stage(the re-ranker mechanism need a post-processing procedure for those samples generated from the AR model, energy scores need to be further calculated by the EBM in their frameworks).
>
>     *p.s.* In principle, the energy-resampling technique can also further improve the performance of our method, just like Residual EBM, see the results in Table1 and Table 7 of our paper. However, it was not the core purpose of our design—-we attempt to benefit the AR model by the proposed training strategy directly.

---

> ### Author Response · Authors · 2022-12-01
> **Sincerely Looking Forward to Your Reply**
>
> Dear Reviewer,
>
> Your suggestions and comments are so valuable. To resolve your concerns, we have made our responses as follows two weeks ago. We sincerely look forward to your reply to our response, and we are open to any discussion to improve our paper.
>
> Best wishes,
>
> The authors.

---

> > ### Author Response · Authors · 2022-12-13
> > **A friendly reminder**
> >
> > Hi Reviewer,
> >
> >    I just want to send a kind reminder. Has our response addressed your concerns? The reviewer discussion period is nearly ended, it would be good if you can have a look at our response. Thank you!
> >
> >   Best wishes,
> >    The authors

---

### Official Review · Reviewer_nMbm · 2022-10-21

**Confidence:** 3
**Correctness:** 4
**Technical Novelty And Significance:** 3
**Empirical Novelty And Significance:** 3
**Recommendation:** 8

**Clarity, Quality, Novelty And Reproducibility:**

**Clarity** — The paper is generally clear, with the basic aspects of the approach well-described in mathematical expressions / equations. I appreciated the algorithm box, which clarified the training procedure. Results tables are also well-formatted and explained in the text.

**Quality** — The paper is fairly high-quality. The authors take a set of principled ideas stemming from energy-based models and apply them to autoregressive models to solve some well-known issues with the latter. Experiments on multiple datasets with relevant baselines demonstrate that the approach is beneficial, and further experiments demonstrate improvements with respect to exposure bias and coherence. In total, the paper tackles known challenges using principled ideas and clearly demonstrates an improvement in relevant empirical settings.

**Novelty** — The paper is somewhat novel. While many of the main ideas (treating non-energy-based models as energy-based models, contrastive divergence, importance weighting) are established to varying degrees, combining these components to train autoregressive models is a novel contribution, as far as I am aware.

**Reproducibility** — I am unsure of the reproducibility of the results, but I suspect that they are reproducible. The authors demonstrate their proposed approach on multiple datasets in various settings, although it’s unclear whether multiple random seeds are used. Generally, their proposed approach outperforms the relevant baselines, with more targeted analyses demonstrating these benefits. While I am unsure of how difficult it would be to reproduce these results, their consistency leads me to believe that they are.

**Strength And Weaknesses:**

**Strengths**

- **Relatively simple/general technique.** The basic idea of the proposed technique is to identify the softmax logits of an autoregressive model as an energy function, allowing them to replace the typical maximum log-likelihood training procedure with a contrastive-divergence-based training procedure. This is reminiscent of previous methods that have interpreted classifiers as energy-based models (e.g., Grathwohl et al., 2020). In this way, the model is explicitly trained by decreasing the likelihood of generated samples, hopefully reducing the effect of “exposure bias.” Notably, this procedure can, in principle, be applied to any discrete autoregressive model, making this a rather general technique.

- **Experiments on multiple datasets with relevant baselines.** The authors demonstrate their proposed approach on language modeling and translation tasks, with three language modeling datasets and one translation dataset with six translation pairs. The authors compare their approach with the baseline autoregressive model, as well as Residual EBM, another method that combines autoregressive and energy-based models. The proposed approach generally compares favorably.

- **Clear description, for the most part.** The descriptions within the paper are generally clear, with both text and mathematical terms well-defined. One way to improve the clarity of the paper even further would be to include some form of diagram.


**Weaknesses**

- **Additional computational overhead for sampling.** One downside of employing a contrastive divergence-based training method is that it requires sampling during training. Sampling from autoregressive models is typically quite costly, and I would guess that this significantly impacts training speed (although this isn’t discussed in the main paper).

- **Only applicable (at least currently) to discrete models.** The current method, as presented, is only applicable to discrete autoregressive models. While this encompasses a wide variety of applications, the authors have not demonstrated how to apply this technique to continuous autoregressive models, e.g., those that output Gaussian densities. In this sense, the title is a bit misleading, as the technique does not apply to all ‘autoregressive models.’

- **Only demonstrated on text data, i.e., no images or audio, etc.** The authors only demonstrate their technique on text data (language modeling and translation, i.e., conditional language modeling). However, even with discrete autoregressive models, this technique could be readily applied to image and perhaps audio modeling. Currently, the paper may only appeal to a subset of the machine learning community focused on language, but with these added application areas, the paper may receive wider attention.

**Summary Of The Paper:**

The authors propose to use the softmax logits in the output of discrete autoregressive models as an energy function, converting an autoregressive model into an energy-based model. They then use contrastive divergence training, using an importance weighted estimate of the “negative phase” gradient. Through experiments on language modeling and translation tasks, they show that this results in more coherent and in-distribution samples, as compared with autoregressive baselines.

**Summary Of The Review:**

The paper presents a fairly general and sound technique for improving autoregressive models by training them as energy-based models. Results are demonstrated on several text datasets, comparing with relevant baselines. In total, the generality of the idea, combined with the paper’s convincing empirical demonstration, leads me to suggest acceptance.

---

> ### Author Response · Authors · 2022-11-15
> **Response to Reviewer nMbm, part[2/2]**
>
> [**Only applicable (at least currently) to discrete models**]
>
> It is a very valuable question. To be honest, we did not consider “strict” continuous generative models in this paper, since people usually apply autoregressive models to continuous data by discretizing those data first in practice, see pixelRNN[5] for example. However, theoretically speaking, the current model can be extended to models with continuous output distribution seamlessly, because energy-based models can definitely model continuous densities. We can just replace the softmax using an EBM.
>
> [**Only demonstrated on text data, i.e., no images or audio, etc**]
>
> Thanks for the advice, it is true that our method empirically focuses on the text data and we do plan to conduct further work in other ML fields apart from the text data. The simple experiments added in appendix C.2 show the application of our E-Forcing training over the image generation tasks. Specifically, we apply E-Forcing to Pixel-CNN [3] and its variant Gated Pixel-CNN[4]. Experiments are carried out on the MNIST and CIFAR-10 datasets. This table summarizes the quantitative results measured by per-pixel negative log-likelihood (NLL). We can see that with the help of our E-Forcing, both the Pixel-CNN and the Gated Pixel-CNN can obtain improvements in all datasets (0.17 $\rightarrow$ 0.15 and 3.14 $\rightarrow$ 3.07 for Pixel-CNN on MNIST and CIFAR10 respectively and 0.14 $\rightarrow$ 0.12 and 3.03 $\rightarrow$ 2.97 for Gated Pixel-CNN on MNIST and CIFAR10 respectively). This is further evidence in favor of our energy-based learning objective for improving autoregressive models. We will carry out more thorough experiments on image data as well as other modalities such as audios, which will be the target of our next work.
>
> [1] Bhattacharyya S, Rooshenas A, Naskar S, Sun S, Iyyer M, McCallum A. Energy-based reranking: Improving neural machine translation using energy-based models. ACL 2021.
>
> [2] Yuntian Deng, Anton Bakhtin, Myle Ott, Arthur Szlam, and Marc’Aurelio Ranzato. Residual energy-based models for text generation. ICLR 2020
>
> [3] Aaron Van Oord, Nal Kalchbrenner, and Koray Kavukcuoglu. Pixel recurrent neural networks. ICML 2016
>
> [4] Aaron van den Oord, Nal Kalchbrenner, Oriol Vinyals, Lasse Espeholt, Alex Graves, and Koray Kavukcuoglu. Conditional image generation with pixel-cnn decoders. NIPS 2016
>
> [5] Will Grathwohl, Kuan-Chieh Wang, Jorn-Henrik Jacobsen, David Duvenaud, Mohammad Norouzi, and Kevin Swersky. Your classifier is secretly an energy based model and you should treat it like one. ICLR 2020
>
> [6] Yilun Xu, Yang Song, Sahaj Garg, Linyuan Gong, Rui Shu, Aditya Grover, and Stefano Ermon. Anytime sampling for autoregressive models via ordered autoencoding. ICLR 2021

---

> > ### Comment · Reviewer_nMbm · 2022-12-05
> > **Response to authors**
> >
> > Thank you for your detailed reply, and thank you, in particular, for running additional experiments on image datasets. This is a strong paper, and I feel that it should be accepted.

---

> ### Author Response · Authors · 2022-11-15
> **Response to Reviewer nMbm, part[1/2]**
>
> Dear Reviewer nMbm, we deeply appreciate your valuable comments and advise. Below are our responses to your concerns. If you have any problems with our response or have more questions about our work, it is welcome to give more comments, and we will try our best to give our response as soon as possible.
>
> [**Additional computational overhead for sampling**]
>
> - First and foremost, we admit that the training speed would be affected to some extent compared to the vanilla training via teacher forcing since our E-Forcing has to generate “fake” samples autoregressively during the optimization procedure. *However, one merit of our E-Forcing is that it can benefit the AR model even without additional overhead in the inference stage*. In contrast, other energy-based methods[1, 2] which utilized the EBM as a re-ranker(re-sampler) inevitably introduce additional overheads not only in the training stage(They need to train an independent EBM apart from the trained AR model) but also in the inference stage(the re-ranker mechanism need a post-processing procedure for those samples generated from the AR model, energy scores need to be further calculated by the EBM in their frameworks). After all, in most cases, people care more about inference efficiency instead of training efficiency, especially for those works based on EBMs which are usually notorious for training efficiency[5].
>
>     *p.s.* In principle, the energy-resampling technique can also further improve the performance of our method just like Residual EBM, see the results in Table1 and Table 7 of our paper, however, it was not the core purpose of our design—-we attempt to directly benefit the AR model by the proposed training strategy.
>
> - Second, for the training time, different implementation tricks applied and hardware types used will largely influence the final training time, and different researchers, who hold different numbers of computing resources, hold a distinct tolerance towards the training time required. In general, our method requires about 2.4 times over translation tasks(IWSLT14 dataset) and 3.1 times over language modeling tasks(Wiki103 dataset) than the training time of original AR training (teacher-forcing). Well, it is true that our E-Forcing is not that efficient at the training stage, however, if we compare it with other Energy-based methods, things might be much more tolerable. For example, JEM[5] trained 36 hours on the simple CIFAR 10 dataset while the vanilla training of a classifier usually costs no more than 3 hours(The main reason is due to the rather time-consuming MCMC procedure during each training iteration), Residual EBM and EBR[1, 2] requires no less than 2 times training time compared to vanilla training since they require to train two independent but same size networks, and it can not be accelerated via concurrently training these two networks since their training of EBM require a well-trained AR model so that the training has to carry out sequentially.
>
>     There are several parts of our method, which can influence the final training efficiency. The first one is the max-length allowed for generated sequences during the training via the contrastive divergence method while the second one is the start-epoch of using E-Forcing training. Setting different max-lengths of generated sequences and different start-epoch of using E-Forcing loss can introduce trade-offs between training efficiency and performance (Please refer to Appendix A.2 for detailed setting over different benchmarks).
>
> - Besides, one possible way to accelerate the training efficiency of our methods is to incorporate the idea shown in the work of Xu et al [6], who proposed a unique way that uses autoregressive models in the latent space followed by a decoder that decodes the autoregressively generated latent feature into the original data space, instead of constructing an autoregressive model in the data space. If we can move the autoregressive procedure from the data level to the latent feature level, our E-Forcing training efficiency can correspondingly improve since those samples generated in the latent space are enough in this case.

---

### Official Review · Reviewer_FpT4 · 2022-10-26

**Confidence:** 4
**Correctness:** 3
**Technical Novelty And Significance:** 3
**Empirical Novelty And Significance:** 2
**Recommendation:** 5

**Clarity, Quality, Novelty And Reproducibility:**

The paper is clear and easy to follow. The proposed method is novel although some parts such as using importance sampling for estimating partition function have been introduced before for similar joint distribution form: p(x) = q(x) exp (-E(x)).




**Strength And Weaknesses:**


Strength:
-The paper is well-written and easy to follow.
-The idea of viewing AR models as EBMs is not novel but the provided formulation is neat and interesting.
-I like the extensive study of the proposed approach.

Weakness:
The improvement especially in the translation tasks is not considerable. Other training algorithms on top of ML training of AR can achieve similar improvement, for example, combining it with RL training.
Bhattacharyya et al. (2021) reported a significant improvement over base AR-NMT by using an external EBM defined over the whole sentence (although some of the improvements come from using pre-trained language models).
Also, their training algorithm can directly be applied to AR by viewing E as \sum_i E_i.

Questions:
1) The author claims this approach can help with long-range dependencies, but I am not seeing how defining the energy model using the last softmax can help with that. Is that the effect of weights on training log q(x<k)? Even so, it is not similar to defining an explicit energy model over the whole sentence!

Bhattacharyya S, Rooshenas A, Naskar S, Sun S, Iyyer M, McCallum A. Energy-based reranking: Improving neural machine translation using energy-based models. ACL 2021.


**Summary Of The Paper:**

AR models are very common in many domains, especially language models. Maximum likelihood is the widely used approach for training these models. However, ML training of AR models causes some other issues such as exposure bias. Techniques such as scheduled sampling have been introduced before to address these issues.
This paper introduces a new training algorithm for AR models by viewing it as an energy-based model.
Here the authors define the model probability as $p(x_1, \cdots, x_k) = q(x_{<k}) \frac{e^{-Q(x_k, x_{<k})}}{Z_\theta}$. They defined $Q(x_k, x_{<k})$ using the logits before the softmax of the $k$th token, so $p(x_1, \cdots, x_k) = q(x_i, \cdots, x_k)$ (am I right?).
However instead of ML training which normalize $e^{-Q(x_k, x<k)}$ with $Z_\theta$, they are training $p(x_i, \cdots, x_k)$ using contrastive divergence (CD). This formulation helps to better regulate the distribution on the $k$th softmax since CD training contrast $Q(x_k, x_{<k})$ with a sample from the model (using importance sampling given the introduced form for the energy) rather than the log partition function.
The final loss function is a combination of cross entropy and CD.

The authors study the proposed method for different tasks of language modeling, machine translation, as well as image generation.



**Summary Of The Review:**

The paper is interesting but the experimental results do not show considerable benefits for the approach. I don't think that the proposed training algorithm would be adopted by the community for training AR models.

---

> ### Author Response · Authors · 2022-11-15
> **Response to Reviewer FpT4, part[2/2]**
>
> - **EBR is designed especially for translation tasks.** The objective function of EBR adopts the BELU score (while the BELU score must require paired bilingual data) for optimization, while both our E-Forcing and residual EBM[2] do not. Such specialization will introduce more *inductive bias* (BLEU is just a single objective evaluation metric) in their method design which is good for their purpose of improving translation models but make things unfair to compare our method(as well as Residual EBM) directly with theirs in the translation task since we design a general training method for AR models. We admit that EBR is beneficial for translation tasks (which use BELU as the standard metric), but whether it is suitable for other metrics is still uncertain. While in our experiments, we evaluate translation (using BLEU, in Section 5.2), language modeling tasks (using PPL, in Section 5.1), and Image generation (using NLL, in Appendix C.2). Furthermore, we also investigate ROUGE, METEOR metrics for translation tasks in Appendix C.7. These results also demonstrate the generalization of our model in different tasks and metrics.
> - **EBR method used an extra network which is a pre-trained Bert model**. A pre-trained energy estimator might bring some additional improvement to the final performance. However, the usage of additional, rich pretrain data makes a direct comparison with our model inappropriate.
>
> Finally, though the direct comparison between general AR training methods and specialized re-ranker-based translators is unfair due to the factors analyzed above, we can still observe some useful information by comparing their performance. Specifically, [1] has reported results of their EBR method as well as the method similar to Residual EBM(which they denoted as NCE-EBR) on the IWSLT14 benchmark while we also reported in Table 2 of our paper regarding the IWSLT14 benchmark. We show those results over EN$\rightarrow$DE, DE$\rightarrow$EN pairs, which are all reported by both papers based on the vanilla transformer architecture, as follows.
>
> | Methods | DE →EN | EN→ DE  |
> | --- | --- | --- |
> | NCE-EBR | 34.47 | 28.22 |
> | Marginal-EBR | 35.68 | 30.82 |
> | Conditional-EBR | 37.58 | 30.97 |
> | E-Forcing | 35.36 | 29.11 |
>
> From the results shown above (since we did not comprehensively proofread the experimental settings between the two papers, the analysis following is not formal), we can observe that our E-Forcing achieved a slightly better performance compared to the NCE-EBR method as well as a slightly worse performance compared to the Marginal-EBR, and the conditional-EBR(Joint EBM) achieved a much better performance compared to others. According to the analysis above, we think such results are actually in line with expectations since EBR used an extra pre-trained EBM model which directly adapted to the BLEU scores, which is a strong inductive bias towards the translation benchmark.
>
> **[Why our E-Forcing training method can help with long-range dependence]**
>
> Well, generally speaking, it is due to the fact that we attempt to model the joint distribution $p_d(x_1,\dots,x_k)$  at each time step $k$ so that E-Forcing can assess the sequence up to the current time step as a whole. Specifically, since we use the logits before the softmax layer to represent the $\phi(\cdot)$ function in $p_\theta(x_1, \dots, x_k) = q_\theta(x_1, \dots, x_{k-1})\frac{e^{-\phi_\theta(x_1, \cdots, x_k)}}{Z}$, the energy-based learning will increase the value of pre-softmax logits for real data(more closed to real distribution $p_d$) while decreases it for fake data (away from $p_d$).
>
> [1] Bhattacharyya S, Rooshenas A, Naskar S, Sun S, Iyyer M, McCallum A. Energy-based reranking: Improving neural machine translation using energy-based models. ACL 2021.
>
> [2] Yuntian Deng, Anton Bakhtin, Myle Ott, Arthur Szlam, and Marc’Aurelio Ranzato. Residual energy-based models for text generation. ICLR 2020
>
> [3] Aditya Grover, Jiaming Song, Ashish Kapoor, Kenneth Tran, Alekh Agarwal, Eric J Horvitz, and Stefano Ermon.  NIPS 2019.
>
> [4] Julian Salazar, Davis Liang, Toan Q. Nguyen, Katrin Kirchhoff. Masked Language Model Scoring. ACL 2020.

---

> > ### Comment · Reviewer_FpT4 · 2022-12-05
> > **Reaction to Author Response**
> >
> > Thank you for your clarification. I believe I have ignored the dependence of $\phi$ on $x_{<k}$ regardless of the fact that you are using the logits for $x_k$ only. However, the degree of freedom for modeling the joint distribution is restricted to the last logits, which may describe the limited improvement in performance.
> >
> > The reported comparison to EBR here is misleading. Looks like you have ignored the base model. The base model that you used for reporting 35.36 on DE->EN (which is incorrectly labeled here as EN->DE) is 35.10, while EBR's is 33.87.
> > E-Enforcing improvement for all reported IWSLT baselines is less than 1 BLEU score, which is very insignificant and is my main objection to the importance of the proposed method.
> >
> > I will keep my rating as before.

---

> > > ### Author Response · Authors · 2022-12-06
> > > **Response to Reviewer FpT4's New Comments**
> > >
> > > Dear reviewer FpT4,
> > >
> > > Very thanks for your reaction to our response. We are glad to see that some of your concern has been resolved, while there is one concern about the performance gain in translation tasks remains. Regarding this concern, there are several points we want to further explain:
> > >
> > > - We acknowledge that the reported comparison to EBR is a bit involved and may cause confusion. Regarding the performance of the base model, we want to point out that the 35.10 BLEU score of our baseline model is achieved by applying several engineering techniques, such as scheduled sampling. The baseline model reported by the EBR method did not use these techniques, and thus is weaker than our baseline.  The result of our implementation of teacher forcing equipped with a bunch of standard tricks (beam search and label smoothing) is 34.61, which is very close to the 34.44 BLEU score on the [leaderboard](https://paperswithcode.com/sota/machine-translation-on-iwslt2014-german) regarding [1]. In contrast, the 33.87 BLEU score reported by EBR, which also applied the beam search and label smoothing techniques, is lower than 34.44. This may be due to some implementation details, we don't know. But we kept our baseline to be as strong as possible to facilitate fair and honest comparison.
> > > - A major portion of EBR's performance gain comes from pretraining, which utilized a large chunk of additional data that is unavailable for our model. Thus, it is unfair to compare EBR's gain directly with our model.
> > > - In academia, we don't have enough computational resources to run hyperparameter sweeping for large datasets such as machine translation, so our performance gain on this task is achieved almost without any tuning efforts, we believe this is the main reason why the gains are relatively small on this task. Besides, our E-Forcing is a general training method for ARGM, which has shown its effectiveness in translation, language modeling, and image generation(see appendix). So while we agree with you that the limited gain on MT is a weakness of this paper, we still believe the contribution of our work is solid.
> > > - Compared to typical works trying to address same problems [2][3][4], our performance gain is quite reasonable and significant.
> > >
> > > We sincerely wish you can take these factors into consideration and reconsider your overall rating. We are very appreciative of your time and the help offered for our paper's improvement.
> > >
> > > [1] Ashish Vaswani, Noam Shazeer, Niki Parmar, Jakob Uszkoreit, Llion Jones, Aidan N. Gomez, Lukasz Kaiser, and Illia Polosukhin. Attention is all you need. NIPS, 2017.
> > >
> > > [2] Tsvetomila Mihaylova and Andr´e F. T. Martins. Scheduled sampling for transformers. ACL 2019
> > >
> > > [3] Yuntian Deng, Anton Bakhtin, Myle Ott, Arthur Szlam, and Marc’Aurelio Ranzato. Residual energy-based models for text generation. ICLR 2020
> > >
> > > [4] Kaitao Song, Xu Tan, and Jianfeng Lu. Neural machine translation with error correction. IJCAI, 2020.

---

> ### Author Response · Authors · 2022-11-15
> **Response to Reviewer FpT4, part[1/2]**
>
> Dear Reviewer FpT4, sincerely thanks for your valuable comments. Below are our responses to your comments and questions. If you have any further problems with our response or have more questions about our work, it is welcome to give more comments. We will try our best to give our response as soon as possible.
>
> **[ $p_\theta$  is not equal to $q_\theta$]**
>
> Thanks for your question. In our definition, $p_\theta$ is defined as $p_\theta(x_1, \dots, x_k) = q_\theta(x_1, \dots, x_{k-1})\frac{e^{-\phi_\theta(x_1, \cdots, x_k)}}{Z}$ (where $\phi(\cdot)$ is $Q(\cdot)$ in your comments, we use $\phi$ to denote it henceforth in order to align with the notation in our paper). According to this definition, it is true that $p_\theta(x_k|x_1,...x_{k-1}) = q_\theta(x_k|x_1,...x_{k-1})$, but it does not mean that  $p_\theta(x_1,\dots, x_k)$  is equal to $q_\theta(x_1,\dots, x_k)$. This is because the marginal probability $p_\theta(x_1,\dots, x_{k-1})$ will be reweighed by logits w.r.t. the time step k. It worths noting that the partition function (i.e., the normalization term) $Z$ is equal to $ E_{q_\theta}[ \sum_{x_k}  e^{-\phi_{\theta}(x_k, x_{< k})}] $ instead of $\sum_{x_k}  e^{-\phi_{\theta}(x_k, x_{< k})}$, which might cause the confusion.
>
> **[The comparison to the EBR method and the concern about the performance of NMT task]**
>
> Thank you for pointing out our missing reference.  We will add EBR [1] to the related works and discuss the differences between EBR and our work in our paper. But we also want to point out there are several fundamental points with respect to the difference between EBR and our method, making it slightly unfair and inappropriate to compare these two works directly:
>
> - **Our method is designed with a fundamentally different goal than EBR**. The main goal of our model is to improve the training of autoregressive models. The EBR method and residual EBM [2], however, treat the AR model as fixed and train an independent energy-based model $E_\theta$ to reweight the samples. These methods are mainly focused on improving performance in terms of reranking, which can be considered post-processing methods [4]. We want to highlight that our method’s design is orthogonal with the reranking-based methods: E-Forcing is not a technique to improve a trained fixed AR model, it is a new energy-based training method for the AR model, and the goal is to improve the AR model itself via energy-based training. In other words, we view the AR model itself as an EBM while EBR, as well as Residual EBM, used EBM as a re-ranker apart from the AR model. Therefore, it is unfair to directly compare our model with reranking-based methods.

---

> ### Author Response · Authors · 2022-12-01
> **Sincerely Looking Forward to Your Reply**
>
> Dear Reviewer,
>
> Your suggestions and comments are so valuable. To resolve your concerns, we have made our responses as follows two weeks ago. We sincerely look forward to your reply to our response, and we are open to any discussion to improve our paper.
>
> Best wishes,
>
> The authors.

---

### Author Response · Authors · 2022-11-22
**Looking Forward to Further Discussions**

Dear reviewers,

We want to appreciate again for your valuable time and your insightful comments. We sincerely look forward to your reply to our response to let us know if we have resolved your concerns. If you have any further problems with our response or have more questions about our work, it is welcome to give more comments. We will try our best to give our response as soon as possible.

Best regards.

The authors

---

### Decision · Program_Chairs · 2023-01-20

**Decision:**

Reject

**Justification For Why Not Higher Score:**

The novelty is not high enough.

**Justification For Why Not Lower Score:**

The paper is properly done and shows merit of the proposed method. One can always argue about the significanceof the results - will they carry over to LLMs for example, but this is for future research to find out.

**Metareview: Summary, Strengths And Weaknesses:**

The paper is about using concepts from energy-based models to improve the training of auto-regressive generative models. Specifically, an energy-based model is defined in eq 3 for the joint distribution up to step k written as a product of the auto-regressive model up step k-1 times a Boltzmann term divided by a normalizer. The learning objective is the likelihood plus the KL between the energy based and auto-regressive model.

Promising results on language modeling and machine translation are presented.

(For this AC, it is not super clear why this approach per se should solve fundamental problems with purely auto-regressive modeling. The empirical results support this so as many other things in DL one perhaps just has to accept that.)

The reviewers are a bit mixed but overall leaning towards accept. The mixed opinions could possible have been resolved in a meeting, but unfortunately due to a too tight program this meeting was not organised. This AC's fault.

This is a bordeline paper and one can argue either way for accept or reject. Since the reviews are mixed and the average score is below the typical for accepted papers, rejection is recommended. The reviews are constructive and can help improve the paper for the next conference.